# KSHV activates unfolded protein response sensors but suppresses downstream transcriptional responses to support lytic replication

**Benjamin P. Johnston**[1,2], **Eric S. Pringle**[1,2], **Craig McCormick**[1,2]*

**1** Department of Microbiology & Immunology, Dalhousie University, Halifax, Nova Scotia, Canada, **2** Beatrice Hunter Cancer Research Institute, Halifax, Nova Scotia, Canada

* craig.mccormick@dal.ca

**Data Availability Statement:** All relevant data are within the manuscript and its Supporting Information files.

## Abstract

Herpesviruses usurp host cell protein synthesis machinery to convert viral mRNAs into proteins, and the endoplasmic reticulum (ER) to ensure proper folding, post-translational modification and trafficking of secreted and transmembrane viral proteins. Overloading ER folding capacity activates the unfolded protein response (UPR), whereby sensor proteins ATF6, PERK and IRE1 initiate a stress-mitigating transcription program that accelerates catabolism of misfolded proteins while increasing ER folding capacity. Kaposi's sarcoma-associated herpesvirus (KSHV) can be reactivated from latency by chemical induction of ER stress, which causes accumulation of the XBP1s transcription factor that transactivates the viral *RTA* lytic switch gene. The presence of XBP1s-responsive elements in the *RTA* promoter suggests that KSHV evolved a mechanism to respond to ER stress. Here, we report that ATF6, PERK and IRE1 were activated upon reactivation from latency and required for efficient KSHV lytic replication; genetic or pharmacologic inhibition of each UPR sensor diminished virion production. Despite UPR sensor activation during KSHV lytic replication, downstream UPR transcriptional responses were restricted; 1) ATF6 was cleaved to activate the ATF6(N) transcription factor but ATF6(N)-responsive genes were not transcribed; 2) PERK phosphorylated eIF2α but ATF4 did not accumulate; 3) IRE1 caused XBP1 mRNA splicing, but XBP1s protein did not accumulate and XBP1s-responsive genes were not transcribed. Ectopic expression of the KSHV host shutoff protein SOX did not affect UPR gene expression, suggesting that alternative viral mechanisms likely mediate UPR suppression during lytic replication. Complementation of XBP1s deficiency during KSHV lytic replication inhibited virion production in a dose-dependent manner in iSLK.219 cells but not in TREx-BCBL1-RTA cells. However, genetically distinct KSHV virions harvested from these two cell lines were equally susceptible to XBP1s restriction following infection of naïve iSLK cells. This suggests that cell-intrinsic properties of BCBL1 cells may circumvent the antiviral effect of ectopic XBP1s expression. Taken together, these findings indicate that while XBP1s plays an important role in reactivation from latency, it can inhibit virus replication at a later step, which the virus overcomes by preventing its synthesis.

**Funding:** This work was funded by an operating grant to CM from the Canadian Institutes for Health Research (MOP-84554) (http://www.cihr-irsc.gc.ca/e/193.html). The funders had no role in study design, data collection and analysis, decision to publish, or preparation of the manuscript.

**Competing interests:** The authors have declared that no competing interests exist.

These findings suggest that KSHV hijacks UPR sensors to promote efficient viral replication while sustaining ER stress.

## Author summary

Like all viruses, Kaposi's sarcoma-associated herpesvirus (KSHV) uses cellular machinery to create viral proteins. Some of these proteins are folded and modified in the endoplasmic reticulum (ER) and traverse the cellular secretory apparatus. Exceeding ER protein folding capacity activates the unfolded protein response (UPR), which resolves ER stress by putting the brakes on protein synthesis and turning on stress-mitigating genes. We show that KSHV replication activates the three cellular proteins that sense ER stress, which are each required to support efficient viral replication. By contrast, KSHV blocks the UPR gene expression program downstream from each of these activated sensor proteins. The failure to resolve ER stress might normally be expected to put the virus at a disadvantage, but we demonstrate that reversal of this scenario is worse; when we supplement infected epithelial cells with the UPR transcription factor XBP1s to artificially stimulate the production of UPR-responsive gene products, virus replication is blocked at a late stage and very few viruses are released from infected cells. Taken together, these observations suggest that KSHV requires UPR sensor protein activation to replicate but has dramatically altered the outcome to prevent the synthesis of new UPR proteins and sustain stress in the ER compartment.

## Introduction

Secreted and transmembrane proteins are synthesized in the endoplasmic reticulum (ER), where they are folded by chaperone proteins and modified by glycosyltransferases and protein disulfide isomerases. Demands on the protein folding machinery that exceed ER folding capacity cause the accumulation of misfolded proteins and trigger ER stress [1]. This accrual of misfolded proteins activates the unfolded protein response (UPR) to mitigate the stress [2–4]. The UPR resolves ER stress by transiently attenuating translation, increasing synthesis of folding machinery, increasing lipid biogenesis to expand ER surface area, and degrading misfolded proteins in a process called ER-associated degradation (ERAD). Thus, the UPR adapts the levels of ER-associated biosynthetic machinery to meet demands on the system; however, if proteostasis is not re-established, the UPR switches from an adaptive to an apoptotic response.

The UPR is coordinated by three transmembrane sensor proteins that sample the ER lumen; activated transcription factor-6 (ATF6), protein kinase R (PKR)-like endoplasmic reticulum kinase (PERK) and inositol-requiring enzyme 1 (IRE1). These sensor proteins are maintained in an inactive state by association of their luminal domains with the ER chaperone BiP [5]. In response to ER stress, BiP is mobilized to participate in re-folding reactions in the ER, releasing UPR sensors from their repressed state [6]. Together these three UPR sensors coordinate complementary aspects of an ER stress-mitigating gene expression program.

ATF6 is an ER-localized type II transmembrane protein. Detection of unfolded proteins in the ER lumen causes ATF6 to traffic to the Golgi apparatus, where it is cleaved by Golgi-resident site-1 protease (S1P) and site-2 protease (S2P) enzymes [7,8], which releases the amino-terminal ATF6(N) fragment into the cytosol. ATF6(N) is a basic leucine zipper (bZIP) transcription factor that translocates to the nucleus and transactivates genes encoding chaperones, foldases and lipogenesis factors.

PERK is an ER-localized type I transmembrane kinase. ER stress causes displacement of inhibitory BiP proteins from PERK, which triggers dimerization and trans-autophosphorylation [9]. Active PERK phosphorylates serine 51 of eIF2α, which increases eIF2α affinity for its guanine exchange factor eIF2B [10,11]. This binding depletes the small pool of eIF2B, thereby inhibiting replenishment of the eIF2-GTP-Met-tRNA$^{Meti}$ ternary complex required for translation initiation [12]. Bulk cap-dependent translation is attenuated, while a subset of uORF-containing mRNAs encoding stress response proteins are preferentially translated [13]. Activating transcription factor 4 (ATF4) is chief among these stress response proteins [14]; this bZIP transcription factor translocates to the nucleus and drives the synthesis of gene products that mitigate ER stress by increasing the antioxidant response and a catabolic process known as autophagy [15,16]. ATF4 upregulates protein-phosphatase 1 α (PP1α) cofactor GADD34 (PPP1R15A), which dephosphorylates eIF2α and allows recharging of the ternary complex and resumption of translation [17,18]. This direct control of protein synthesis by stress is known as the integrated stress response (ISR) [19,20]. During chronic or severe ER stress, ATF4 also upregulates the bZIP transcription factor CHOP, which promotes stress-induced cell death by coordinating the expression of genes that promote apoptosis [21–23].

IRE1 is an ER-localized type I transmembrane kinase and endoribonuclease. ER stress causes release of repressive BiP proteins from IRE1, which triggers IRE1 dimerization and trans-autophosphorylation, and stimulates RNase activity. On the cytosolic face of the ER, active IRE1 cleaves a conserved nucleotide sequence in two stem loops in the *X-box binding protein-1 (Xbp1)* mRNA, which excises a 26-nucleotide intron [24–26]. The tRNA ligase RTCB completes this cytosolic mRNA splicing event by re-ligating cleaved *Xbp1* mRNA, which generates a shifted open reading frame that can be translated to produce the active bZIP transcription factor XBP1s [27,28]. XBP1s translocates to the nucleus and drives production of chaperones, lipid synthesis proteins and proteins involved in ERAD [29]. The combined action of these gene products simultaneously increases ER folding capacity and decreases the load via ERAD. In a process called regulated IRE1-dependent decay (RIDD), IRE1 can also cleave select ER-targeted mRNAs with a stem loop that resembles that of *XBP1*, which may serve to attenuate translation [30,31]. One of the first roles ascribed to *XBP1* was its requirement for terminal B cell differentiation into antibody-secreting plasma cells [32], where it expands ER capacity to support antibody secretion [33].

KSHV is a gammaherpesvirus that causes Kaposi's sarcoma (KS), primary effusion lymphoma (PEL) and multicentric Castleman's disease (MCD) [34–36]. Like all herpesviruses, KSHV can establish a quiescent form of infection known as latency in which viral gene expression is severely restricted and the genome is maintained as a nuclear episome. The virus can latently infect a variety of cell types, but it is thought that life-long infection of human hosts is primarily enabled by latent infection of immature B lymphocytes [37], followed by viral reprogramming into an intermediate cell type that resembles a plasma cell precursor [38–40]. Indeed, PEL cells display elevated UPR gene expression consistent with a plasma cell-like phenotype [40].

An essential feature of latency is reversibility, which is required for viral replication and production of viral progeny. The true physiologic cues for KSHV lytic reactivation remain obscure, but *in vitro* studies have implicated ER stress-activated signalling pathways [41–43]. Reactivation from latency requires the immediate-early lytic switch protein replication and transcriptional activator (RTA), a transcription factor that initiates a temporal cascade of gene expression [44,45]. RTA recruits host cell co-factors to transactivate viral early genes required for genome replication [46]. KSHV has usurped the IRE1/XBP1s pathway to regulate the latent/lytic switch. XBP1s binds to canonical response elements in the RTA promoter and drives synthesis of the RTA lytic switch protein required for lytic reactivation [42]. Functional

XBP1s binding motifs have also been found in the promoter for v-IL6 [47], the KSHV homo-log of human IL6, which plays key roles in PEL and MCD [48–50]. Thus, UPR activation causes reactivation from latency and initiation of lytic gene expression. Interestingly, because the XBP1 transcription factor is required for normal B cell differentiation into mature plasma cells [51,52], it is likely that the viral acquisition of XBP1s target sequences hardwires KSHV reactivation from latency to terminal B cell differentiation [42], although the physiological consequences of this link are not known. In sharp contrast to these mechanistic linkages, nothing is known about how ATF6 and PERK impact latent KSHV infection or reactivation from latency.

KSHV encodes structural and non-structural proteins that are folded in the ER and traverse the secretory apparatus. UPR activation during herpesvirus lytic replication has been reported, and there is evidence for UPR sensor engagement by specific gene products, rather than simply by exceeding ER folding capacity [53–55]. For example, herpes simplex virus type 1 (HSV-1) glycoprotein B can bind and suppress PERK activation to promote virus protein production [56]. To better understand how KSHV usurps the UPR, we investigated UPR activation following reactivation from latency in B cell- and epithelial cell-based models. We report that all three proximal UPR sensors are activated following reactivation from latency; ATF6 is cleaved, PERK and eIF2α are phosphorylated, and IRE1 is phosphorylated and *Xbp1* mRNA is spliced. Furthermore, we determined that UPR sensor activation is pro-viral; pharmacologic or genetic inhibition of each UPR sensor diminished virion yield from infected cells. Surprisingly, viral proteins accumulated despite sustained phosphorylation of eIF2α throughout the lytic cycle, suggesting that viral messenger ribonucleoproteins (mRNPs) may have unique properties that ensure priority access to translation machinery during stress. Remarkably, activation of proximal UPR sensors during lytic replication failed to elicit any of the expected downstream effects: ATF6(N) and XBP1-targets genes were not upregulated and ATF4 was not translated. Indeed, despite clear evidence of IRE1 activation and *XBP1* mRNA splicing, XBP1s protein failed to accumulate during KSHV lytic replication. This suggests that the virus requires proximal activation of UPR sensors but prevents downstream UPR transcription required to mitigate stress and restore ER homeostasis. Despite its important role in host shutoff, the KSHV RNA endonuclease SOX did not affect UPR gene expression in ectopic expression models, suggesting that alternative viral mechanisms likely mediate UPR suppression during lytic replication. Remarkably, complementation of XBP1s deficiency during KSHV lytic replication by ectopic expression inhibited the production of infectious virions in the iSLK.219 epithelial cell model, but not in the TREx BCBL1-RTA PEL cell model. Therefore, while XBP1s plays an important role in reactivation from latency, it inhibits later steps in lytic replication in some cells, which the virus overcomes by tempering its synthesis. Taken together, these findings suggest that KSHV hijacks UPR sensors to promote efficient viral replication instead of resolving ER stress.

## Results

### KSHV lytic replication activates IRE1 and PERK but downstream UPR transcription factors XBP1s and ATF4 do not accumulate

ER stress can induce KSHV lytic replication via IRE1 activation and XBP1s-mediated transactivation of the viral *RTA* latent/lytic switch gene [41,42]. However, it is not known whether the burden of synthesizing secreted and transmembrane KSHV proteins causes ER stress over the course of the lytic replication cycle. To test UPR activation during the lytic cycle we used the TREx BCBL1-RTA cell line that expresses RTA from a doxycycline (dox)-regulated promoter to reactivate KSHV from latency [57]. We treated cells with dox for 0, 24, and 48 hours (h) and immunoblotted for UPR proteins to determine their activation state and KSHV proteins to

monitor the progression of the lytic cycle from early (ORF45) to late (ORF65) stages (Fig 1A). As a positive control to ensure that UPR sensors were intact in our system, we also treated latent and lytic cells with the SERCA (sarco/endoplasmic reticulum $Ca^{2+}$-ATPase) inhibitor thapsigargin (Tg) to pharmacologically induce ER stress [58]. After 2 h of Tg treatment of latently infected cells, IRE1α was phosphorylated, as determined by slower electrophoretic mobility, and a semiquantitative *XBP1* RT-PCR splicing assay revealed that the majority of *Xbp1* mRNA was spliced, which enabled translation of XBP1s protein (Fig 1A). Tg activated the ISR in latent cells, as determined by phosphorylation of PERK (also revealed by slower electrophoretic mobility of PERK) and its downstream target eIF2α-Ser51, which promoted translation of ATF4 from a uORF-containing mRNA. Lysates collected at 24 and 48 h post-dox addition displayed strong accumulation of ORF45 and ORF65, which indicated progression through early and late lytic replication, respectively. By 24 h post-dox, IRE1α and PERK pathways were both activated: IRE1 and PERK were phosphorylated, which corresponded to increased spliced *XBP1* mRNA and phospho-eIF2α, respectively. Activation of the IRE1 and PERK pathways persisted until 48 h post-dox. However, PERK activation and eIF2α phosphorylation did not activate the ISR in lytic cells; ATF4 mRNA levels in latent and lytic cells were identical (Fig 1B), there was negligible accumulation of ATF4 protein (Fig 1A), and levels of ATF4-dependent CHOP mRNA did not increase (Fig 1B). Tg treatment of lytic cells at 24 h post-dox caused a strong increase in PERK activation and eIF2α phosphorylation, but once again, ATF4 protein did not accumulate and mRNA levels of the ATF4-target gene CHOP were significantly less than elicited by Tg-treatment of latently-infected cells (Fig 1A and 1B). Taken together, these observations suggest that KSHV lytic replication prevents downstream execution of the ISR when eIF2α is phosphorylated. By 48 h post-dox, Tg treatment was no longer able to stimulate accumulation of phospho-eIF2α, and total PERK levels were diminished compared to earlier stages of replication (Fig 1A). Tg-treated lytic cells also displayed reduced total IRE1α and XBP1s levels, even though *Xbp1* mRNA splicing was similar to Tg-treated latent cells. These data demonstrate that KSHV lytic replication activates IRE1α and PERK UPR sensor proteins but prevents the accumulation of XBP1s and ATF4 transcription factors required for downstream UPR transcriptional responses.

## KSHV lytic replication activates ATF6 but downstream UPR transcriptional responses are inhibited

The third UPR sensor ATF6 is cleaved in the Golgi during ER stress to release its active N-terminal fragment ATF6(N) that translocates to the nucleus and transactivates UPR genes [8]. We observed elevated total endogenous ATF6 levels at 24 h post-dox, which diminished over the following 24 h (Fig 1A). However, the ATF6 antibody that we employed in this study could not detect endogenous ATF6(N) in any samples, including Tg-treated positive controls; this is consistent with previous reports of rapid degradation of the labile ATF6(N) protein [8,59]. To further investigate ATF6 cleavage in this system, we transduced TREx BCBL1-RTA cells with a lentiviral vector encoding HA-epitope-tagged full-length ATF6 (HA-ATF6α-FL) [60]. This ectopic expression system allowed us to monitor cleavage of the ~100 kDa HA-ATF6α-FL precursor into the ~60 kDa HA-ATF6α N-terminal fragment, which was revealed after Tg treatment (Fig 1C, lane 2). Consistent with endogenous ATF6 in lytic cells (Fig 1A), we observed the accumulation of full-length HA-ATF6 at 24 h post-dox followed by return to basal levels by 48 h post-dox. During lytic replication, we also observed the cleavage of HA-ATF6α into the active 60 kDa fragment (Fig 1C, lanes 3–6). Indeed, by 24 h post-dox the levels of HA-ATF6α-N were higher than latently infected cells treated with Tg. Interestingly, ATF6-target genes BiP and HERPUD1 [61–63] were not transactivated during lytic replication, even in

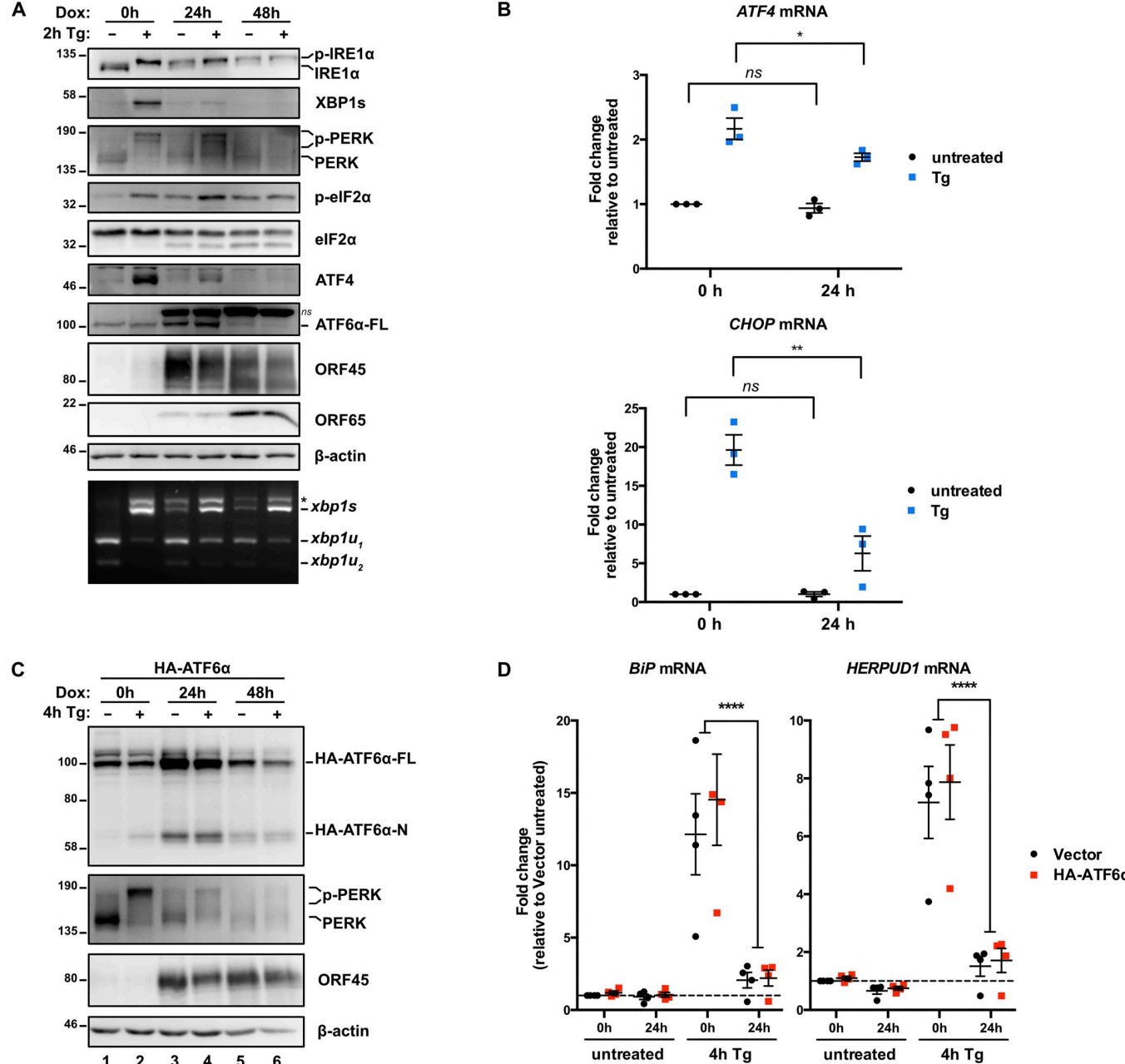

**Fig 1. KSHV lytic replication activates IRE1, PERK and ATF6 but downstream UPR transcription factors are inhibited.** (A) TREx BCBL1-RTA cells were treated with 1 μg/mL dox for 0, 24 and 48 h to induce lytic replication followed by 75 nM Thapsisgargin (Tg) for 2 h prior to harvesting for protein and total RNA. Whole cell lysates were analyzed by immunoblots for UPR markers (IRE1α, XBP1s, PERK, phospho- and total eIF2α, ATF4, and full length ATF6α). Migration shift in PERK and IRE1α immunoblots correspond to phosphorylation. KSHV proteins ORF45 and ORF65 were probed for to indicate induction of early lytic and late lytic, respectively. β-actin was used as a loading control. ns corresponds to an unknown protein species that cross-reacted with ATF6 anti-sera. *Xbp1* mRNA was amplified by RT-PCR, digested with PstI (cleaves unspliced XBP1 isoform only), and separated on SYBR Safe-stained agarose gel. The asterisk (*) corresponds to xbp1u-xbp1s hybrid cDNA. Representative immunoblot and agarose gel of two independent experiments are shown. (B) TREx BCBL1-RTA cells were treated with 1 μg/mL dox for 0 or 24 h to induce lytic replication and 4 h prior to harvesting for total RNA, cells were treated with or without 75 nM Tg. Relative changes in mRNA levels of ATF4 and CHOP were measured by qPCR and calculated using the ΔΔCt method using 18S rRNA as a reference gene. (*, p value < 0.05, **, p value < 0.01) (C) TREx BCBL1-RTA cells were transduced with lentiviral expression vector encoding HA-ATF6α and selected for with 1 μg/mL puromycin. Following selection, cells were treated with 1 μg/mL dox for 0, 24 and 48 h and treated with 75 nM Tg for 4 h prior to harvest. Whole cell lysates were analyzed by immunoblots for HA epitope tag, PERK, ORF45 and β-actin (loading control). Immunoblot shown is representative of two independent experiments. (D) As in (C), HA-ATF6a-transduced TREx BCBL1-RTA were treated with dox for 0 and 24 h and then treated with 75nM Tg for 4 h prior to total RNA isolation. mRNA levels of ATF6α target genes BiP and HERPUD1 were measured by qPCR. Changes in mRNA levels were calculated by the ΔΔCt method and normalized using 18S rRNA as a reference gene. An average of 3 independent experiments are graphed and error bars denote SEM. Two-way ANOVA and a post-hoc multiple comparisons tests were done to determine statistical significance (****, p value < 0.0001).

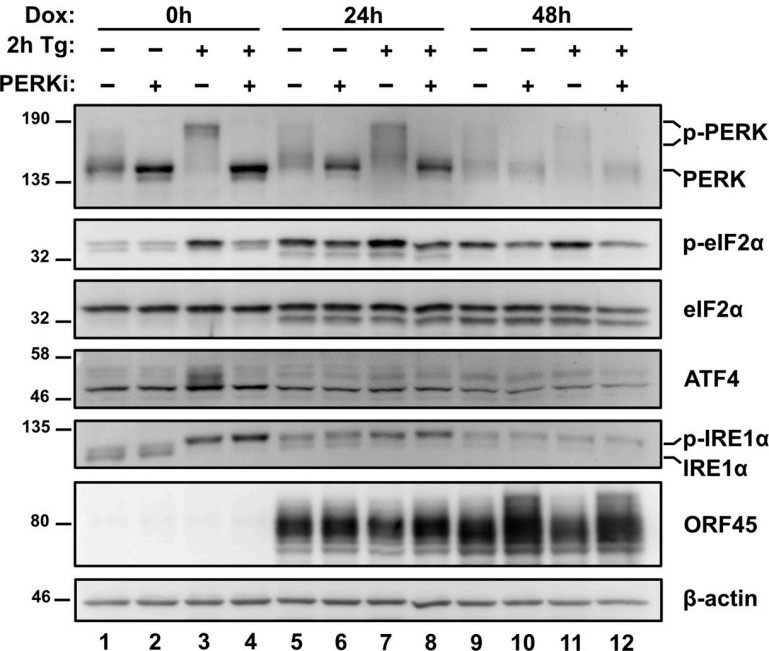

**Fig 2. PERK-dependent eIF2α phosphorylation during the KSHV lytic cycle.** TREx BCBL1-RTA cells were treated
with 1 μg/mL dox -/+ 500 nM of the PERK inhibitor GSK2606414 (PERKi) for 0, 24 and 48 h and treated with or
without 75 nM Thapsisgargin (Tg) for 2 h prior to harvest. Whole cell lysates were analyzed by immunoblots for
PERK, phospho and total eIF2α, ATF4, IRE1α, and ORF45. Migration shift in PERK and IRE1α immunoblots
correspond to phosphorylation. β-actin was used as a loading control. Immunoblots shown are representative of two
independent experiments.

cells treated with Tg (Fig 1D). These data show that ATF6 is proteolytically cleaved during
lytic replication but the transcription factor cannot transactivate canonical target genes.

## PERK-dependent eIF2α phosphorylation during the KSHV lytic cycle

Because eIF2α can be phosphorylated on serine 51 by four possible eIF2α kinases activated by
different stresses (ER stress activates PERK, dsRNA activates PKR, nutrient stress activates
GCN2, oxidative stress activates HRI [9,64–66]), we investigated the specific contribution of
PERK to eIF2α phosphorylation during KSHV lytic replication with the PERK inhibitor
GSK2606414 (PERKi) [67,68], which fully inhibited PERK and eIF2α phosphorylation follow-
ing 2 h Tg treatment (Fig 2, lane 3 vs 4). TREx BCBL1-RTA cells were treated with PERKi con-
current with dox addition. After 24 h and 48 h treatment with dox and PERKi, there was a
reduction in PERK and eIF2α phosphorylation compared to cells treated with dox alone (Fig
2, lanes 5 vs 6 & 9 vs 10), indicating that PERK is involved in phosphorylating eIF2α during
lytic replication. Importantly, PERKi had no appreciable effect on IRE1α activation, indicating
that ER stress is still manifested during KSHV lytic replication when PERK is inhibited, and
PERK inhibition does not heighten this response. Furthermore, lytic cells treated with PERKi
in the presence or absence of Tg also showed increased levels of ORF45 (Fig 2, lanes 9 vs 10 &
lanes 11 vs 12), which suggests that even though ATF4 fails to accumulate, PERK-mediated
phosphorylation of eIF2α may still impede viral protein synthesis to some extent. Finally, the
level of eIF2α phosphorylation following PERKi treatment did not return to baseline levels
observed in latently infected TREx BCBL1-RTA cells, suggesting that other eIF2α kinases
besides PERK may be activated during lytic replication.

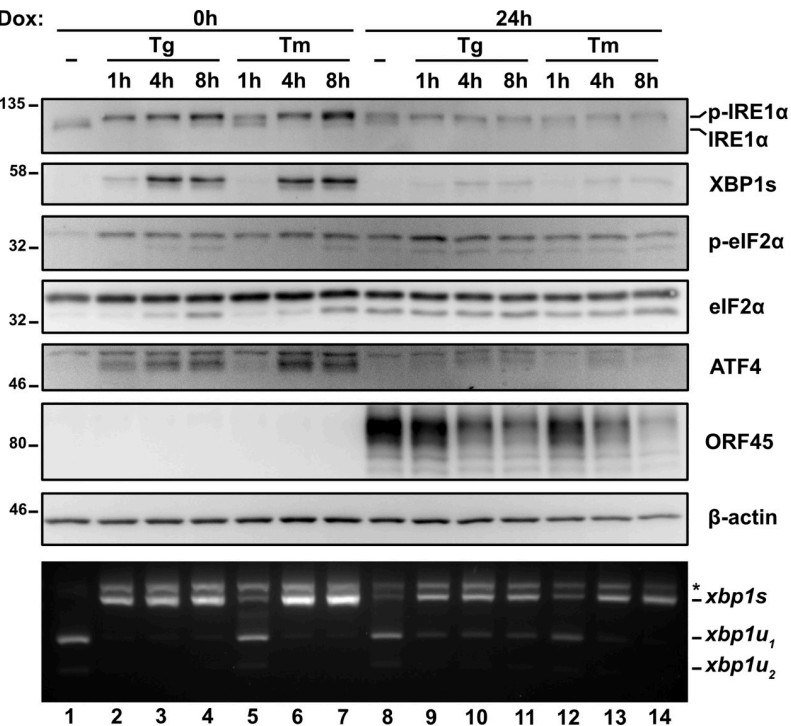

**Fig 3. The type or duration of ER stress does not account for the lack of ATF4 and XBP1s.** TREx BCBL1-RTA cells were pretreated with dox for 0 or 24 h and treated with 75 nM Thapsisgargin (Tg) or 5 μg/mL Tunicamycin (Tm) for 1, 4, or 8 h prior to harvest for either protein or RNA. Whole cell lysates were analyzed by immunoblots for UPR markers (IRE1α, XBP1s, phospho- and total eIF2α, and ATF4). Migration shift in IRE1α immunoblot corresponds to phosphorylation. KSHV protein ORF45 was probed to show lytic replication and β-actin was used as a loading control. *Xbp1* RT-PCR splicing assay was performed as previously indicated. (*) corresponds to xbp1u-xbp1s hybrid cDNA. Immunoblots and agarose gels are representative of two independent experiments.

## KSHV lytic replication suppresses XBP1s and ATF4 accumulation irrespective of the type or duration of ER stress

We previously observed UPR sensor activation during the lytic cycle but failure to accumulate active UPR transcription factors XBP1s and ATF4 following a brief 2 h pulse of Tg (Fig 1A). We confirmed these observations by treating latent and lytic TREx BCBL1-RTA cells with Tg or tunicamycin (Tm), which induces ER stress by blocking protein N-linked glycosylation in the ER [69,70], over a range of incubation times (1, 4 and 8 h). The majority of *Xbp1* mRNA splicing was observed after latent cells were Tg-treated for 1 h or Tm-treated for 4 h (Fig 3); these slower kinetics are consistent with known properties of Tm, which relies on new protein synthesis to trigger ER stress [71]. As expected, in latently infected cells, *Xbp1* mRNA splicing and XBP1s and ATF4 protein accumulation were observed throughout the 8 h course of Tg or Tm treatment. By contrast, during KSHV lytic replication, regardless of duration of Tg or Tm treatment, comparable levels of spliced *Xbp1* mRNA and phospho-eIF2α were detected but XBP1s and ATF4 proteins failed to accumulate. Consistent with previous results, there were slightly higher levels of unspliced *Xbp1* mRNA observed during the lytic cycle that correlated with reduced total IRE1α protein levels, but this was only observed in Tg-treated cells; Tm treatment converted the bulk of the *Xbp1* mRNA pool to the spliced form by 4 h post-treatment, even though IRE1α accumulation was still blocked. Taken together, these observations confirm that XBP1s and ATF4 transcription factors do not accumulate in the KSHV lytic cycle despite robust activation of UPR sensors in response to chemically induced ER stress.

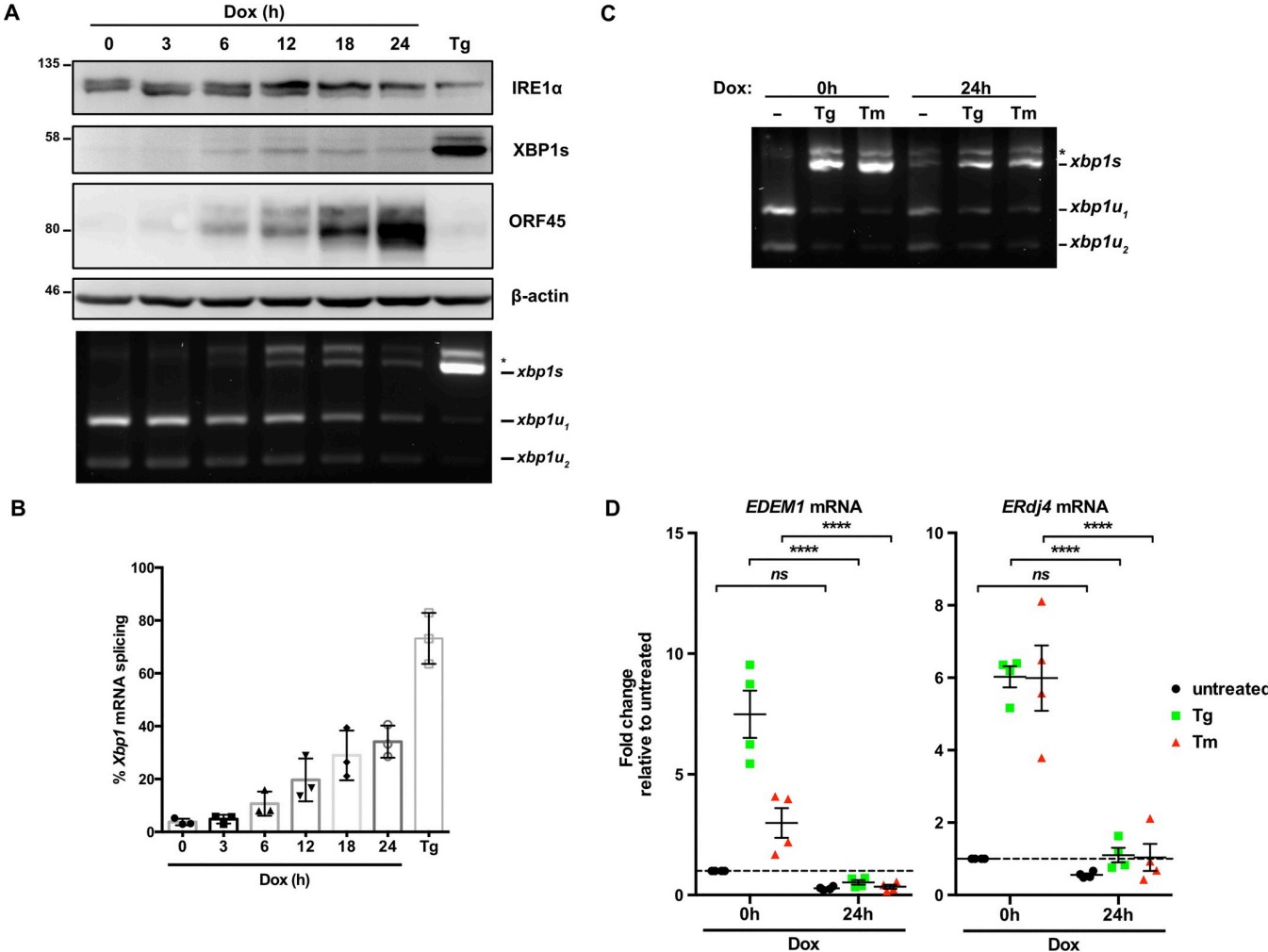

**Fig 4. Low levels of XBP1s induced in lytic are insufficient to upregulate XBP1s-target genes.** (A) TREx BCBL1-RTA cells were treated with 1 μg/mL dox for the indicated times or treated with 75 nM Thapsisgargin (Tg) for 4 h (as a positive control) and harvested for either protein or RNA. Whole cell lysates were analyzed by immunoblots for IRE1α, XBP1s, ORF45 and β-actin (loading control). *Xbp1* RT-PCR splicing assay was performed as previously indicated. (*) corresponds to xbp1u-xbp1s hybrid cDNA. A representative immunoblot and agarose gel for three independent experiments are shown. (B) Densitometry analysis from semi-quantitative *XBP1* RT-PCR splicing assay in (A) was used to calculate the percentage of *XBP1* in the spliced isoform. The mean of three independent experiments are graphed and error bars represent the standard deviation of the mean. (C) TREx BCBL1-RTA cells were treated with 1μg/mL dox for 0 or 24 h and treated with 75 nM Tg or 5 μg/mL Tunicamycin (Tm) for 4 h prior to RNA isolation. *XBP1* mRNA splicing was determined by semi-quantitative RT-PCR splicing assay as previously described. The gel shown is representative of two independent experiments. (D) Total RNA samples from (C) were used to measure the mRNA levels of XBP1s target genes EDEM1 and ERdj4 by qPCR. Changes in mRNA levels were calculated by the ΔΔCt method and normalized using 18S rRNA as a reference gene. An average of 4 independent experiments are graphed and error bars denote SEM. Two-way ANOVA and a post-hoc multiple comparisons tests were done to determine statistical significance (****, *p* value < 0.0001).

## XBP1s target genes are not transactivated by XBP1s during lytic replication

To better understand the kinetics of IRE1 activity during KSHV lytic replication we reactivated TREx BCBL1-RTA cells from latency using dox and monitored IRE1 activity over a 24 h time course. Spliced *XBP1* mRNA and XBP1s protein were detected by 6 h post-dox addition, concomitant with a modest increase in IRE1 phosphorylation, as determined by reduced electrophoretic mobility (Fig 4A). This induction of *Xbp1* splicing corresponded with increased accumulation of the early viral protein ORF45. By 18 h post-dox addition, which coincides

with the beginning of viral genome replication in this model [57], IRE1 phosphorylation, *Xbp1* splicing and XBP1s protein accumulation had peaked, and almost 40% of *Xbp1* mRNA was spliced (Fig 4A and 4B). By contrast, Tg treatment converted nearly the entire pool of *Xbp1* mRNA to the spliced form and caused strong accumulation of XBP1s protein. To determine whether the low levels of XBP1s observed during lytic replication were sufficient to induce synthesis of XBP1s target genes, RNA was harvested from latent and lytic (24 h post-dox addition) TREx BCBL1-RTA cells treated with Tg, Tm or vehicle control; Semiquantitative RT-PCR analysis revealed that *Xbp1* mRNA was efficiently spliced following Tg or Tm treatment, both in latent and lytic samples, while moderate splicing was observed in samples from vehicle-treated control lytic cells (Fig 4C). RT-qPCR was performed to measure relative levels of canonical XBP1s target genes EDEM1 and ERdj4 [72] (Fig 4D); Tg and Tm treatments caused dramatic accumulation of EDEM1 and ERdj4 transcripts in latently infected cells, whereas these transcripts remained at low levels in lytic cells treated with Tg, Tm or mock-treated. Thus, low expression of XBP1s during lytic replication was insufficient to induce synthesis of canonical XBP1s target gene products involved in ER stress mitigation.

## KSHV SOX host shutoff protein is not sufficient to block UPR transcriptional responses

Our observations indicate that KSHV activates UPR sensor proteins while simultaneously suppressing downstream XBP1s, ATF6(N), and ATF4 transcriptional responses. During KSHV lytic replication the viral host shutoff endonuclease SOX targets the majority of host mRNAs for degradation [73,74]. This indirectly causes accumulation of host RNA-binding proteins in the nucleus [75] and reduces the recruitment of RNA polymerase II to host promoters, causing global repression of host transcription [76]. To test whether SOX can prevent accumulation of UPR gene products, we engineered a dox-inducible myc-SOX 293A cell line. Cells were transduced with lentiviral vectors encoding the reverse tetracycline-controlled transactivator rtTA3 and myc-SOX under the control of seven tandem Tet operator (TetO) elements and stable cells were selected with antibiotics. Dox was added to induce SOX expression for 48, 72, and 96 h; cells were treated with 500 nM Tg for 4 h prior to harvesting total RNA at each time point. GAPDH, a known substrate of SOX, was included as a positive control [77]. SOX reduced GAPDH mRNA levels ~ 2-fold, both in the presence and absence of Tg (Fig 5A). However, SOX had little effect on the low steady-state levels of UPR transcripts ERdj4, BiP, or CHOP. Tg-treatment dramatically increased UPR gene expression at each stage of the time-course, which was also largely unaffected by SOX, suggesting that SOX does not impede UPR gene expression. To corroborate these findings, cells bearing dox-inducible wild type SOX or a P176S endonuclease-defective SOX mutant [78] were induced over a 48 h time-course and treated with 500 nM Tg or vehicle control for 4 h prior to harvesting protein lysates. Immunoblotting analysis revealed no change in the accumulation of BiP (ATF6-dependent), XBP1s (IRE1-dependent) or CHOP (PERK-dependent) in response to Tg treatment in the presence of WT or mutant SOX (Fig 5B). Likewise, Tm treatment elicited PERK activation, eIF2$\alpha$ phosphorylation and BiP, XBP1s, ATF4 and CHOP accumulation to a similar extent in the presence of either WT or mutant SOX (Fig 5C). These findings indicate that SOX expression had little impact on UPR signaling irrespective of the mode of ER stress induction and is unlikely to play a significant role in inhibiting UPR transcriptional responses during lytic replication.

## UPR sensor activation supports efficient KSHV replication

We next used a combination of genetic and pharmacologic approaches to inhibit each UPR sensor protein and measure effects on KSHV lytic replication. To investigate the role of ATF6,

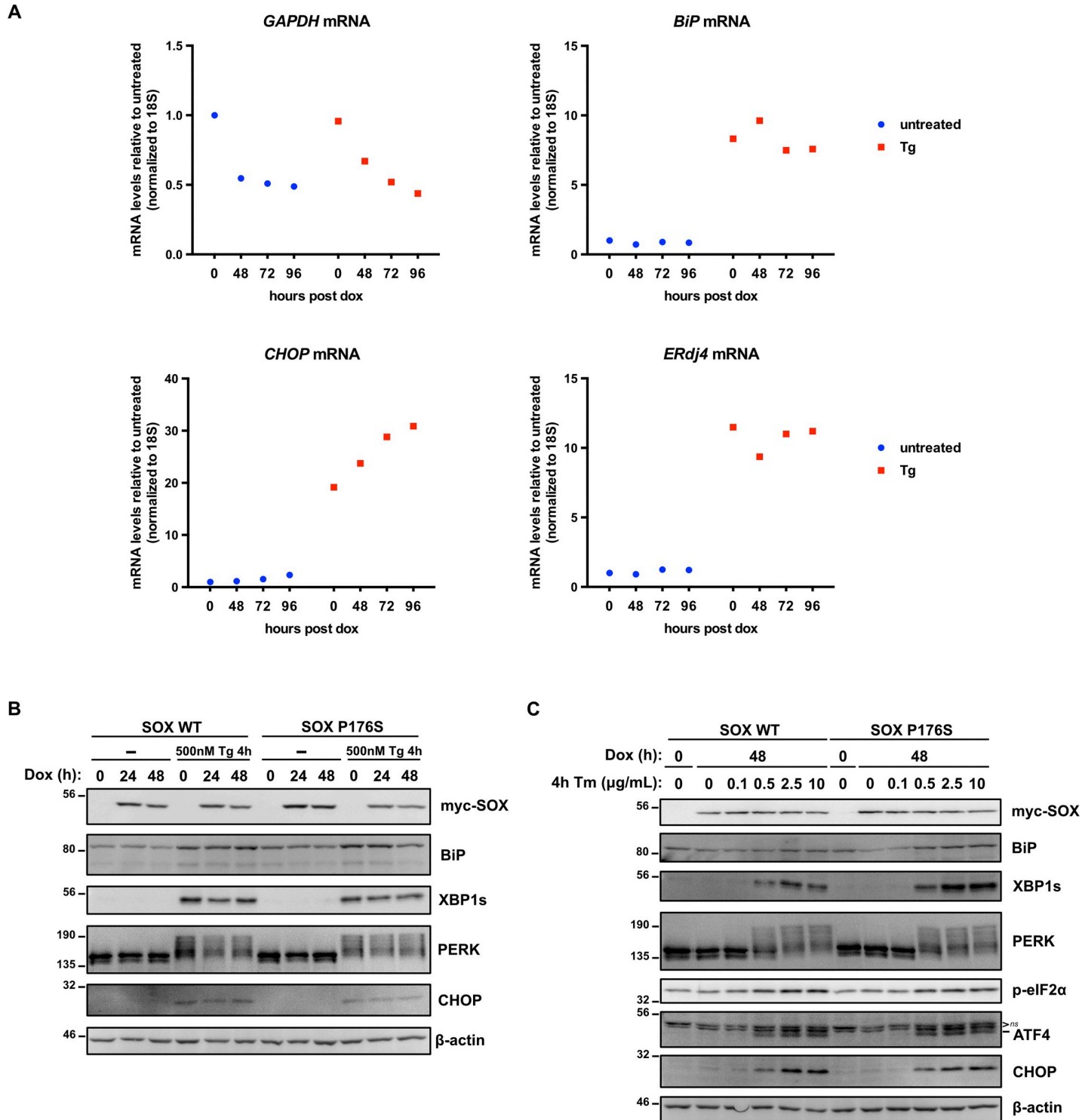

**Fig 5. The KSHV early gene SOX is not sufficient to inhibit the UPR.** (A) 293A Tet-On cells (see Materials and Methods for generation) were transduced with lentiviral expression vectors encoding myc-SOX and selected with blasticidin. SOX expression was induced with 1 μg/ml dox for 0, 48, 72, and 96 h; cells were treated with 500 nM Thapsisgargin (Tg) for 4 h prior to harvesting total RNA. Changes in mRNA levels of ERdj4, BiP, CHOP, and GAPDH were calculated with the ΔΔCt method using 18S rRNA as a reference gene. (B) Dox-inducible myc-SOX wildtype (WT) and myc-SOX P176S point-mutant 293A cells were treated with 1 μg/ml dox for 0, 24, and 48 h and 500 nM Tg was added 4 h prior to harvesting whole cell lysates. Lysates were analyzed by immunoblotting for myc-Sox, BiP, XBP1s, PERK and CHOP. β-actin was used as a loading control. (C) Similar to (B), myc-SOX WT or myc-SOX P176S expression was induced with dox for 48 h and treated with increasing concentrations of Tm at 4 h prior to harvesting total cell lysates. Immunoblot analysis of myc-SOX and UPR substrates BiP, XBP1s, PERK, p-eIF2α, ATF4 (ns, denotes non-specific protein bands), and CHOP. β-actin was used as a loading control. Experiments in A, B, and C were conducted once.

we silenced ATF6α expression in TREx BCBL-RTA cells with shRNAs (Fig 6A) and collected cell supernatants from ATF6-silenced or control cells at 48 h post-dox addition. Cell supernatants were processed to measure relative levels of released capsid-protected viral genomes by qPCR. ATF6 knockdown reduced virus titer by ~50% compared to cells transduced with non-targeting shRNA (Fig 6B). To further confirm a positive role for ATF6 cleavage in KSHV virion production, we treated TREx BCBL1-RTA cells with the ATF6 inhibitor Ceapin-A7 or the S1P inhibitor PF 429242, which selectively inhibited Tg-induced ATF6-target gene BiP and had no effect on XBP1s-target gene ERdj4 (Fig 6C and 6D). Inhibition of ATF6 with Ceapin-A7 and S1P inhibitor inhibited KSHV production in TREx-BCBL1-RTA cells almost 2-fold (Fig 6E). We corroborated these findings in the dox-inducible iSLK.219 cell model that produces recombinant KSHV virions harbouring a GFP transgene. iSLK.219 cells were treated with Ceapin-A7 or S1P inhibitor at the time of dox addition and cell supernatants were harvested 96 h later, serially diluted, and titered on naive monolayers of 293A cells by flow cytometry. Like in TREx BCBL1-RTA cells, inhibition of ATF6 reduced virus titers from iSLK.219 cells (Fig 6F). Interestingly, while Ceapin-A7 had a similar effect on titer in the two cell models, S1P inhibition in iSLK.219 cells caused an ~100-fold decrease in virus titre, indicating that S1P may have an ATF6-independent role during lytic replication in iSLK.219 cells.

To determine whether activation of PERK or downstream engagement of the ISR are important for viral replication TREx BCBL1-RTA cells were treated with the selective PERK inhibitor GSK2606414 (PERKi) or the ISR inhibitor ISRIB, a small molecule that blocks phospho-eIF2α-mediated inhibition of translation by maintaining active eIF2B [79,80]. PERKi and ISRIB each inhibited viral particle release by ~50% (Fig 6G).

To determine if IRE1 RNase activity is required for efficient viral replication TREx BCBL1-RTA cells were treated with the IRE1 inhibitor 4μ8c [81], which inhibited virus release in a dose-dependent manner (Fig 7A). We corroborated these findings in the dox-inducible iSLK.219 cell model [82]; as in the TREx BCBL1-RTA cell model, higher doses of 4μ8c inhibited virion production from iSLK.219 cells, with statistically significant inhibition achieved at the 25 μM dose (Fig 7B). To confirm a role for IRE1 in viral replication, we inhibited IRE1α expression in iSLK.219 cells via RNA silencing. Cells transduced with IRE1α-targeting shRNAs or non-targeting controls were treated with dox for 96 h, and cell supernatants were once again collected to titer GFP-expressing KSHV virions by flow cytometry. iSLK.219 cells bearing IRE1α shRNAs inhibited release of infectious virions by more than two-fold compared to cells transduced with non-targeting shRNAs (Fig 7C and 7D). Taken together, these data suggest that all three sensors of the UPR are important for robust virus replication.

## Ectopic XBP1s expression inhibits release of infectious KSHV virions in a cell type- and dose-dependent manner

Our studies to this point suggest that efficient lytic replication depends on activation of all three UPR sensor proteins, but the cell fails to produce UPR transcription factors and downstream transcriptional responses required to mitigate ER stress. This suggests that KSHV may re-dedicate UPR sensors for a new purpose rather than resolving ER stress, and that sustained, low-level ER stress may not impede viral replication. We also found it puzzling that IRE1 RNase activity was required for efficient lytic replication but XBP1s could not transactivate XBP1s-target genes, including RTA, during the lytic cycle. For these reasons, we hypothesized that XBP1s accumulation may negatively impact KSHV replication. To complement the XBP1s deficiency in the KSHV lytic cycle we overexpressed XBP1s using dox-inducible lentiviral vectors. We engineered myc-tagged XBP1s to be weakly expressed under the control of a single tet operator (TetO) element, or strongly expressed under the control of seven tandem

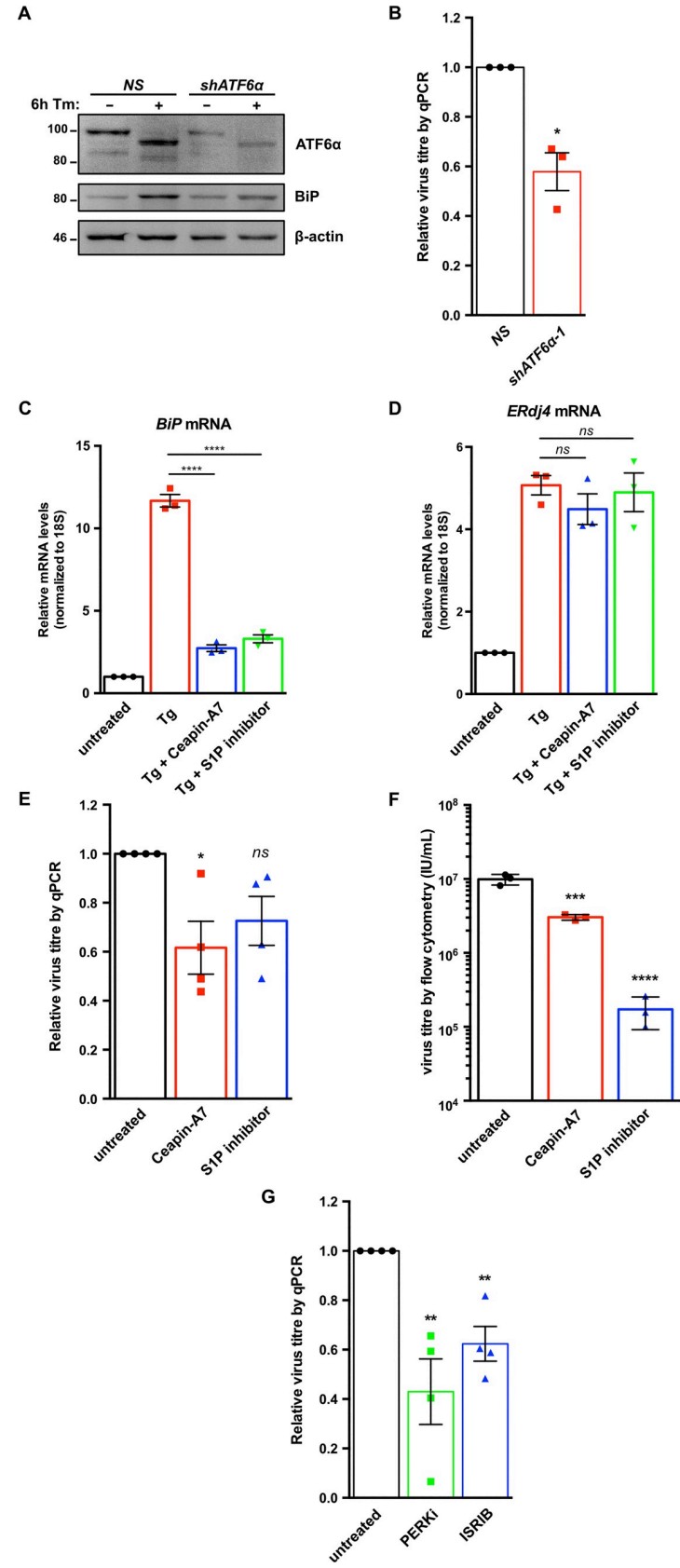

**Fig 6. ATF6α and PERK support robust KSHV replication.** (A) TREx BCBL1-RTA cells were transduced with lentivirus expressing either non-targeting shRNA (NS) or shRNA against ATF6 and selected for with 1 μg/mL puromycin. Following selection, cells were treated with 5 ug/mL Tm for 6 h and whole cell lysates were analyzed by immunoblots for ATF6 and BiP, a transcriptional target of ATF6. (B) As in (A), cells were treated with 1 μg/mL dox for 48 h and virus-protected genomic DNA from cell supernatants were column purified for determining virus titer by qPCR using primers against ORF26. Firefly luciferase plasmid DNA was added during DNA purification for normalization using the ΔΔCt method. (C and D) TREx BCBL1-RTA cells were pre-treated with 10 μM Ceapin-A7 (ATF6 inhibitor) or 10 μM S1P inhibitor (PF 429242) for 20 h followed by 75 nM Thapsisgargin (Tg) for 4 h and then harvested for total RNA. Relative mRNA levels of (C) ATF6-target gene BiP and (D) XBP1-target gene ERdj4 were measured by qPCR, respectively. 18S rRNA was used as a reference gene and changes in mRNA levels were calculated by the ΔΔCt method. (E) TREx BCBL1-RTA cells were treated with 1 μg/mL dox and 10 μM of either Ceapin-A7 or S1P inhibitor PF 429242 for 48 h and virus-protected genomic DNA from cell supernatants was measured by qPCR to determine changes in virus titer using the ΔΔCt method by normalizing to *luc2* DNA levels. (F) iSLK.219 cells were treated with 1 μg/mL dox with or without 10 μM Ceapin-A7 or S1P inhibitor for 96 h and virus-containing supernatants were serially diluted and spinfected onto a monolayer of 293A cells. GFP-positive cells (infected) were quantified by flow cytometry the following day and virus titer (IU/mL) was calculated as described in the Materials & Methods. (G) TREx BCBL1-RTA cells were treated with 1 μg/mL dox for 48 h with or without 500 nM PERKi GSK2606414 or 250 nM ISRIB, and virus titer was measured by qPCR as in (B). Data are represented as 3 (B, C, D, and F) or 4 (E and G) independent experiments and error bars denote SEM. One-way ANOVA and a post-hoc multiple comparisons tests were done to determine statistical significance. (*, p value < 0.05; **, p value < 0.01; ***, p value < 0.001; ****, p value < 0.0001).

TetO elements (7xTetO). We transduced iSLK.219 cells with these constructs or an empty control vector and selected stable cells with blasticidin. With the addition of dox, XBP1s and RTA were concurrently expressed from dox-inducible promoters. To demonstrate that the ectopic XBP1s was functional we measured mRNA levels of the XBP1s-target gene *ERdj4* and observed that it is upregulated in cells expressing XBP1s from the 7xTetO compared to empty vector control and peaks at 24 h post-dox addition (Fig 8A). These cells also displayed higher levels of mRNA and protein for RTA and the RTA target gene ORF45 by 24 h post-dox treatment; and by 48 h, mRNA encoding the late viral protein K8.1 was markedly increased compared to controls (Fig 8A and 8B), suggesting accelerated viral genome replication. Indeed, intracellular levels of viral genomes at 96 h post-dox were two-fold higher in XBP1s-overexpressing cells compared to empty vector (Fig 8C). We harvested cell supernatants at 48, 72, and 96 h post-dox and measured virion titer as previously described. Surprisingly, despite accelerated viral gene expression and genome replication in XBP1s-overexpressing cells, there was a dramatic, dose-dependent reduction in virion production by 72 and 96 h post-dox compared to empty vector control (Fig 8D). There was also a corresponding 20-fold decrease in release of viral particles by XBP1s-overexpressing cells compared to controls, as measured by qPCR for capsid-protected viral genomic DNA (Fig 8E). Thus, while ectopic XBP1s expression promotes KSHV lytic gene expression and genome replication, it prevents efficient release of infectious progeny.

Since XBP1s transactivates the *RTA* promoter, we hypothesized that this defect in virion production could be a negative consequence of *RTA* hyper-activation. We observed a decrease in the accumulation of capsid proteins ORF26 and ORF65 (Fig 8B), which we speculated could result from RTA hyper-activation and negatively impact late stages of replication [83]. However, experiments with the weaker 1xTetO-XBP1s construct revealed a 2-fold decrease in virion release at 96 h post-dox (Fig 8D) without affecting RTA, ORF26 and ORF65 protein accumulation (Fig 8B). To confirm that the diminished production of infectious virions from XBP1s-overexpressing cells is not due to enhanced RTA expression, we also overexpressed RTA in parallel from a 7xTetO in iSLK.219 cells, such that dox addition causes RTA expression by two dox-responsive promoters. KSHV from iSLK.219 cells also express monomeric red fluorescent protein (mRFP) from the viral lytic PAN promoter and can be used to monitor virus reactivation [84]. At 48 h post-dox addition, the levels of mRFP were similar between

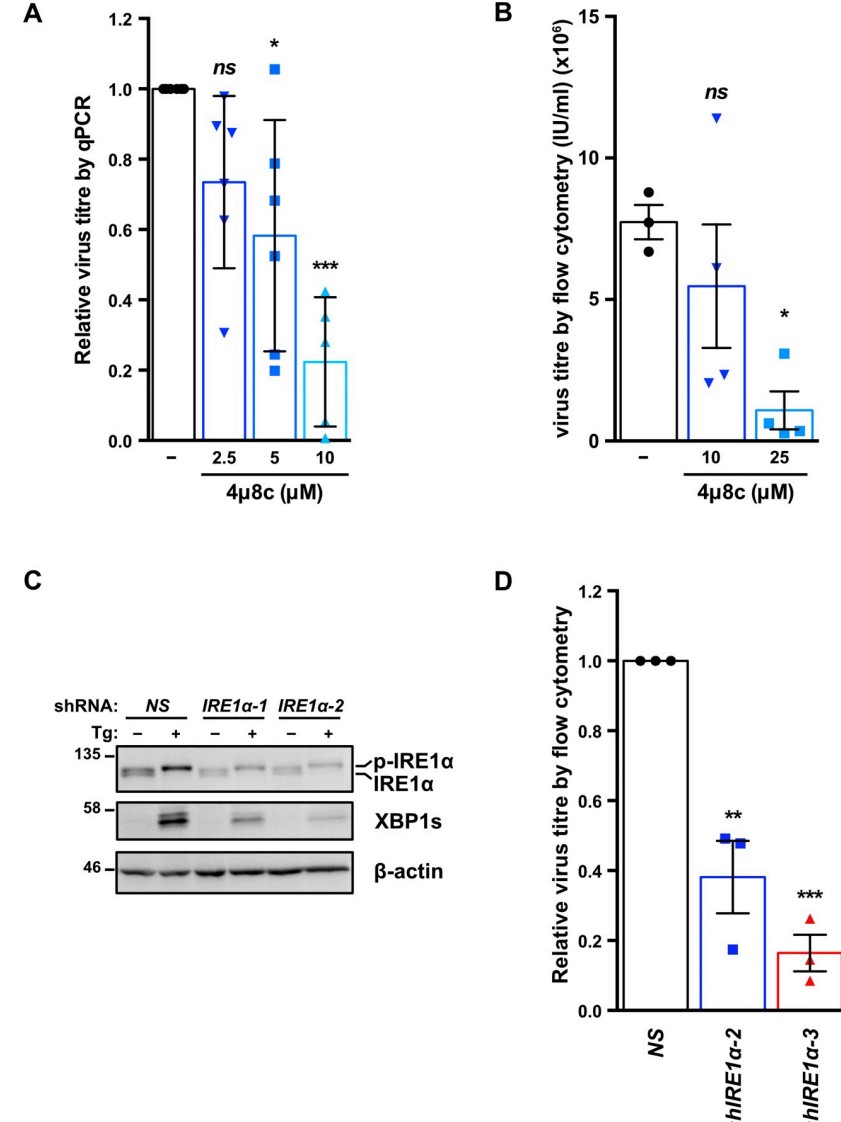

**Fig 7. IRE1 is required for efficient KSHV replication.** (A) TREx BCBL1-RTA cells were treated with 1 μg/mL dox for 48 h with or without increasing concentrations of IRE1 inhibitor 4μ8c and DNase I-protected genomic DNA from cell supernatants was measured by qPCR to determine changes in virus titer using the ΔΔCt method by normalizing to *luc2* DNA levels. The mean of four independent experiments are graphed -/+ SEM. (B) iSLK.219 cells were treated with 1 μg/mL dox with or without 10 or 25 μM 4μ8c for 96 h and virus-containing supernatants were serially diluted and spinfected onto a monolayer of 293A cells. GFP-positive cells (infected) were quantified by flow cytometry the following day and virus titer (IU/mL) was calculated as described in the Materials & Methods. The mean of four independent experiments are graphed -/+ SEM. (C) iSLK.219 cells were transduced with two different pLKO.1-blast shRNA lentiviruses targeting IRE1α or a non-targeting control and selected for with blasticidin. Following selection, cells were treated with 150 nM Thapsisgargin (Tg) for 4 h and harvested for immunoblot analysis to confirm IRE1α knockdown. The immunoblot shown is representative of two independent experiments performed. (D) Lentivirus transduced iSLK.219 cells from (C), were treated with dox for 96 h and virus titer was determined by flow cytometry as previously described. The data are represented as the change in virus titer relative to the non-targeting shRNA control sample and the mean of three independent experiments are graphed -/+ SEM. One-way ANOVA and multiple comparisons test were done to determine statistical significance in (A), (B), and (D). (*ns*, not statistically significant; *, $p$ value $< 0.05$; **, $p$ value $< 0.01$; ***, $p$ value $< 0.001$).

XBP1s and RTA-expressing cells and noticeably greater than that of the empty vector control (Fig 8F). We harvested virus-containing supernatants at 24, 48, 72, and 96 h post-dox and

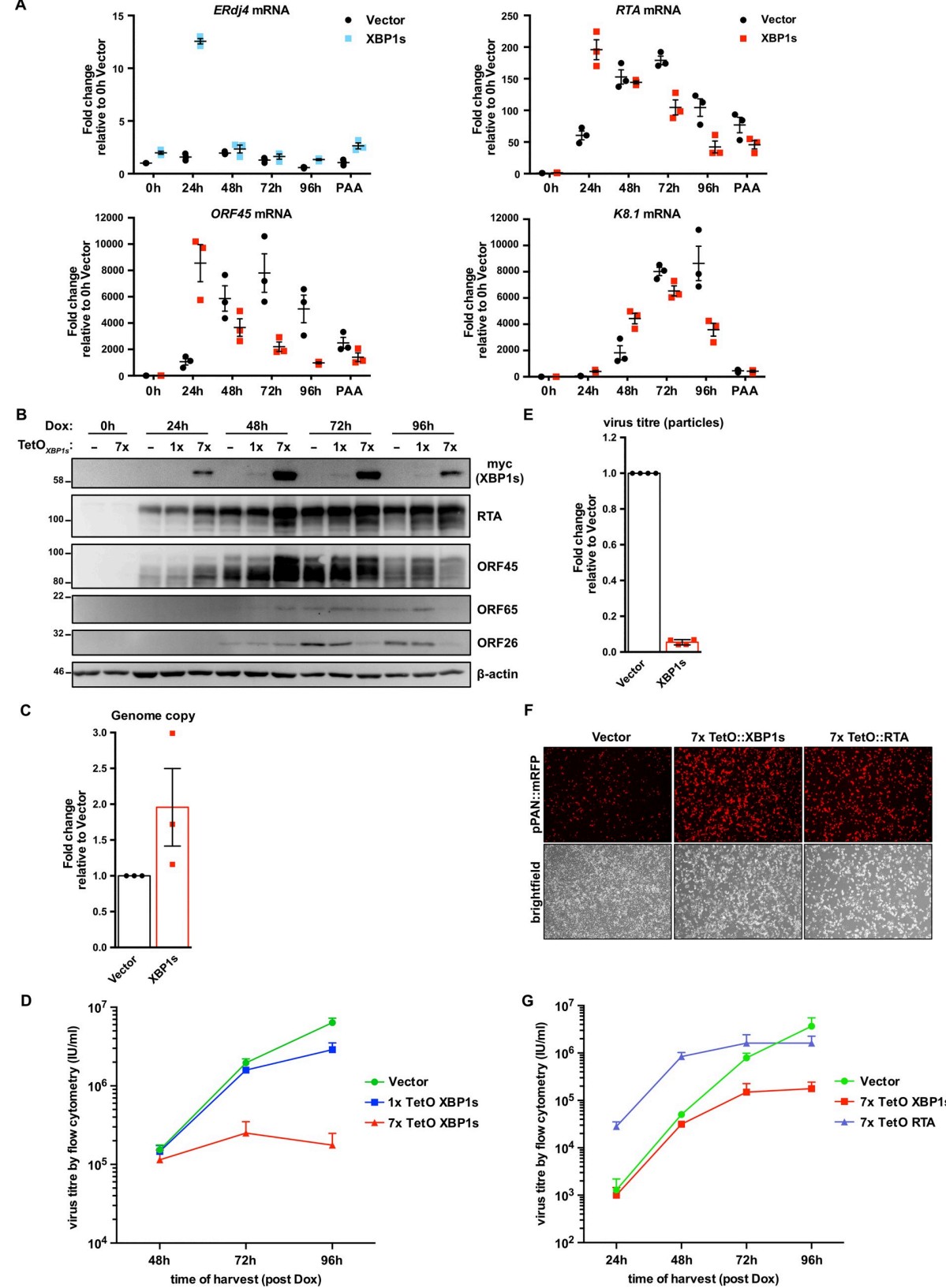

**Fig 8. XBP1s overexpression inhibits KSHV replication at a late stage.** (A) iSLK.219 cells were transduced with lentiviral dox-inducible expression vectors encoding myc-XBP1s whose expression is driven from 7x tandem tet operator (TetO) for robust gene expression. Following blasticidin expression, lytic replication was induced with dox for 0, 24, 48, 72, and 96 h or 96 h with 500 nM PAA (to inhibit genome replication) and harvested for total RNA. mRNA levels of XBP1s-target gene ERdj4 and viral genes RTA (immediate early gene [IE]), ORF45 (early gene [E]), and K8.1 (late gene [L]) were measured by qPCR using 18S rRNA as a reference gene for normalization. An average of 3 independent experiments are graphed and error bars denote SEM. (B) iSLK.219 cells were transduced with lentiviral dox-inducible expression vectors encoding myc-XBP1s whose expression is driven from either 1x TetO (weak expression) or 7x TetO (strong expression) and treated with dox for the indicated times and harvested for total cell lysates. Immunoblots were done for myc-epitope tag (XBP1s), and the viral proteins RTA (IE), ORF45 (E), ORF65 (L), and ORF26 (L). β-actin was used as a loading control. The presented immunoblots are representative of 2 independent experiments. (C) 7xTetO-myc-XBP1s and vector transduced iSLK.219 cells were treated with dox for 96 h and intracellular DNA purified. qPCR against ORF26 DNA was done to measure the relative change in viral genome replication using the ΔΔCt method and normalized to β-actin DNA. Values are the mean of 3 independent experiments -/+ SEM. (D) Virus-containing supernatants from (C) were serially diluted and spinfected onto a monolayer of 293A cells. GFP-positive cells were quantified by flow cytometry the following day and used to calculate virus titer (IU/mL). The values are the mean virus titer of 4 independent experiments -/+ SEM. (E) 7xTetO-myc-XBP1s and vector transduced iSLK.219 cells were treated with dox for 96 h and DNase-protected genomic DNA from supernatants were column purified. Firefly luciferase plasmid DNA was added during DNA purification to allow for normalization. The relative change in virus titer (infectious and non-infectious) was quantified by qPCR using ORF26 primers and normalized to luciferase DNA using the ΔΔCt method. Values are the mean of 4 independent experiments -/+ SEM. (F and G) iSLK.219 cells were transduced with lentiviral expression vectors encoding either dox-inducible 7xTetO-myc-XBP1s or 7xTetO-FLAG-RTA. Following blasticidin selection, lytic replication was induced with dox for (F) 48 h and fluorescence microscopy was used to image RFP-positive cells (cells undergoing lytic replication) or for (G) 24, 48, 72, and 96 h and virus supernatants were harvested to measure titer by flow cytometry. Values are an average of 3 independent experiments -/+ SEM.

measured virion release by flow cytometry following infection of a naïve 293A monolayer (Fig 8G). At 24h, when virion production is negligible in empty vector- and XBP1s-expressing cells, there is a significantly higher level of virions produced by 7xTetO RTA-expressing cells which continues up until 72 h. At 96 h post-dox, virion production from the 7xTetO RTA-expressing cells hits a plateau. Here again, in agreement with our previous observations, virion production from 7xTetO XBP1s-expressing cells is comparable to empty vector control after 48 h but by 96 h post-dox virion production is ~20-fold lower. These data demonstrate that the dramatic reduction in virions produced by XBP1-overexpressing cells is not due to RTA hyper-activation.

Our findings indicate that despite the important role that XBP1s plays in reactivation from latency, ectopic expression of XBP1s suppresses virion production in iSLK.219 cells. To corroborate these observations, we transduced iSLK.219 cells with increasing concentrations of lentiviral vectors that constitutively express myc-tagged XBP1s or FLAG-tagged RTA from a CMV promoter. To ensure that there were would be an equivalent level of reactivation between cells constitutively expressing RTA or XBP1s, we chose virus dilutions that resulted in a similar range of mRFP levels (indicating lytic cycle initiation) as monitored by fluorescence microscopy (Fig 9A). Six days after cell transduction, virus-containing supernatants were harvested, and infectious virions were enumerated by infecting naive 293A cells and detecting GFP positive cells by flow cytometry; as a positive control, we harvested virus from untransduced iSLK.219 cells treated with dox for 72h (Fig 9B). Here, we observed an expected dose-dependent increase in virion release from RTA-transduced iSLK.219 cells. By contrast, increasing levels of XBP1s did not increase virion production by these cells, and the highest virion yield was almost 100-fold lower than the corresponding highest yield from RTA-transduced cells or dox-treated iSLK.219 cells. In RTA-transduced cells, RTA and ORF45 protein levels increase in step with increasing fluorescent signal of the mRFP reporter, but surprisingly, in XBP1-expressing cells, increasing XBP1s expression had a nominal impact on RTA and ORF45 protein accumulation (Fig 9C). However, we observed that escalating doses of ectopic XBP1s altered the electrophoretic mobility of ORF45 and prevented accumulation of the late protein ORF65. These findings confirm that XBP1s prevents release of KSHV virions in the iSLK.219 cell model, but do not pinpoint the defect in replication.

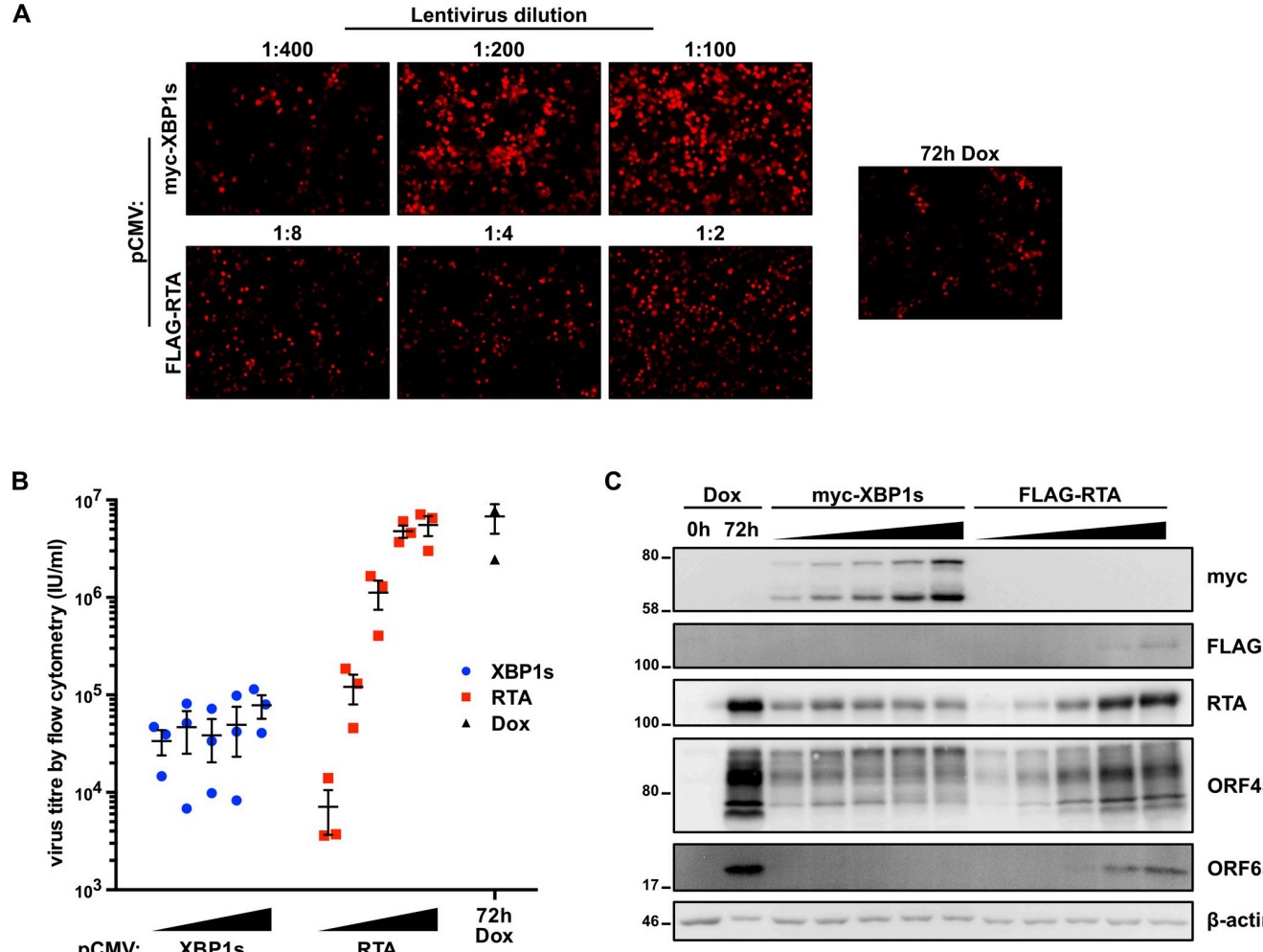

**Fig 9. XBP1s ectopic expression inhibits KSHV replication in iSLK.219 cells.** (A,B, and C) iSLK.219 cells were transduced with increasing concentrations of CMV-driven lentiviral expression vectors encoding myc-XBP1s (LV dilutions: 1:400, 1:200, 1:100, 1:50, 1:25) or FLAG-RTA (LV dilutions: 1:32, 1:16, 1:8, 1:4, 1:2) for 6 days or treated with dox for 72 h (as a positive control) and (A) a subset of the lentiviral dilutions, cells were imaged for RFP-positive cells by fluorescence microscopy; (B) virus-containing supernatants were harvested for measuring virus titer by flow cytometry; and (C) cell lysates were analyzed by immunoblots for myc-epitope tag (XBP1s), and the viral proteins RTA, ORF45, and ORF65. β-actin was used as a loading control.

To determine if this inhibition of virion production by XBP1s is observed in other cell lines, we also transduced TREx BCBL1-RTA cells with increasing concentrations of lentiviral vectors expressing either CMV-driven myc-tagged XBP1s or FLAG-tagged RTA. 48 h after transduction, immunoblot analysis revealed that contrary to the situation in iSLK.219 cells, ectopic expression of XBP1s in TREx BCBL1-RTA cells caused dose-dependent increase in RTA, ORF45, and ORF65 (Fig 10A). Likewise, XBP1s caused a dose-dependent increase in virus production at 72 h post-transduction (Fig 10B). Indeed, cells transduced with the highest concentration of myc-XBP1s lentiviral vector produced as much virus as cells transduced with the highest concentration of FLAG-RTA lentiviral vector or control TREx BCBL1-RTA cells reactivated with dox alone for 48 h. Taken together, these observations suggest that XBP1s induction of the lytic cycle can stimulate a productive infection in TREx BCBL1-RTA cells or an abortive infection in iSLK.219 cells, and that the outcome is likely determined during later stages of replication.

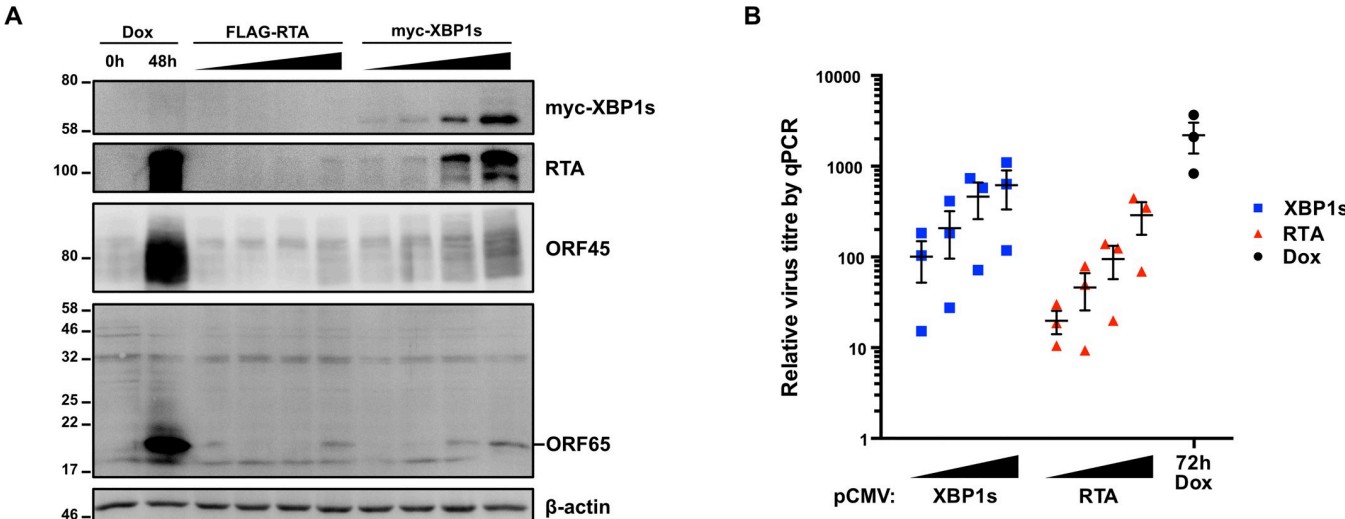

**Fig 10. XBP1s ectopic expression does not inhibit KSHV replication in TREx BCBL1-RTA cells.** (A) TREx BCBL1-RTA cells were treated with 1 μg/mL dox or transduced with increasing concentrations of lentiviral vectors encoding CMV-driven expression of FLAG-RTA (lentivirus dilutions: 1:16, 1:8, 1:4, and 1:2) or myc-XBP1s (lentivirus dilutions: 1:32, 1:16, 1:8, and 1:4) for 48 h. Cell lysates were harvested for immunoblot analysis for myc-epitope tag (XBP1s) and the viral proteins RTA, ORF45, and ORF65. β-actin was used as a loading control. The immunoblots shown are representative of two independent experiments. (B) TREx BCBL1-RTA cells were treated with dox or transduced with the same lentivirus concentrations of FLAG-RTA or myc-XBP1s as in (A). 72 h post-transduction or post-dox, the supernatant was harvested and DNase I- protected viral genomic DNA was measured by qPCR using primers against ORF26. Firefly luciferase plasmid DNA was added during DNA purification for normalization using the ΔΔCt method. The mean of 3 independent experiments is shown and the error bars correspond to the SEM.

The iSLK.219 cells harbour a genetically-modified KSHV (rKSHV.219) derived from the JSC-1 parental strain [84], whereas the TREx BCBL1-RTA cells carry the BCBL1 KSHV strain. These KSHV strains are highly similar, with minor differences evident in internal repeat regions and the K1 gene. To determine whether these subtle differences in KSHV strains determine susceptibility to XBP1s restriction, we transduced naïve iSLK cells with lentiviruses expressing myc-tagged XBP1s or empty vector control, and then infected them with virus-containing supernatant harvested from lytic iSLK.219 or TREx BCBL1-RTA cells. These cells were immediately treated with dox to stimulate RTA transgene expression and bypass latency establishment, spurring lytic KSHV replication. By 4 days post-infection (dpi) we observed KSHV replication and plaque formation in cells transduced with the empty lentiviral control vector which was enhanced by dox treatment (Fig 11A). Plaque formation was most evident in cell monolayers infected with the TREx-BCBL1-RTA-derived virus. By contrast, iSLK cells that expressed myc-XBP1s showed signs of increased lytic reactivation (higher number of RFP-positive cells in rKSHV.219-infected cell monolayers and increased cell detachment in BCBL1-infected cell monolayers), but plaques did not form. This suggests that XBP1s can block plaque formation by both KSHV strains. To quantify this effect, we measured levels of intact progeny viral particles in cell supernatants harvested at 4 dpi by qPCR amplification of capsid-protected genomes, as described above. We observed that ectopic XBP1s expression dramatically reduced the production of both KSHV strains (Fig 11B). This decrease in viral particle production by myc-XBP1s-expressing cells following *de novo* infection correlated with decreased levels of RTA and the early protein ORF57 at 4 dpi, which likely results from a blockade in multi-round replication (Fig 11C). These findings indicate that the JSC-1 and BCBL1 strains of KSHV are equally susceptible to the antiviral effects of XBP1s during lytic replication in epithelial cells.

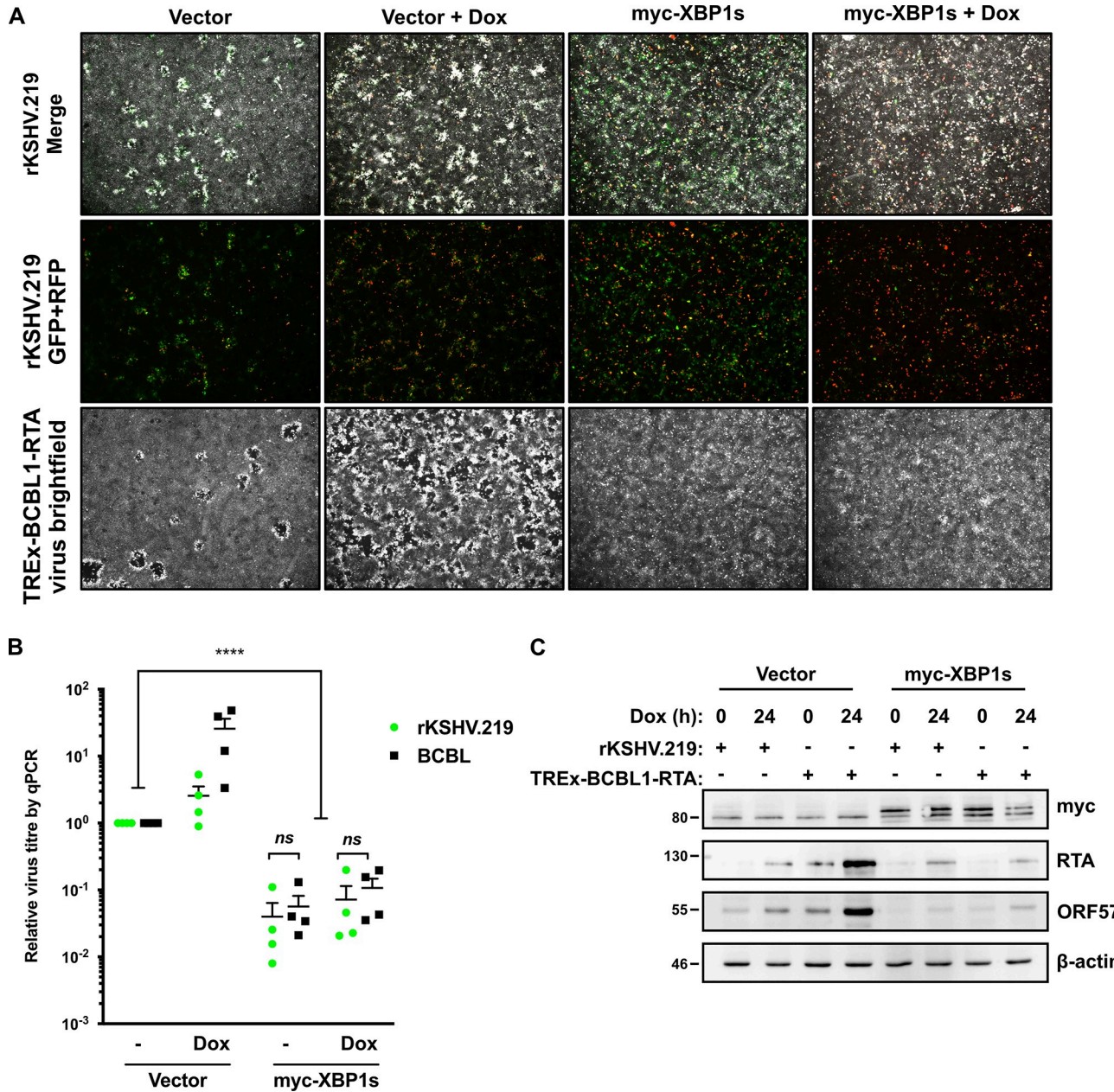

**Fig 11. XBP1s ectopic expression inhibits virion production of both TREx-BCBL1-RTA-derived virus and rKSHV.219 in *de novo* infection.** (A, B, and C) iSLK cells were transduced with CMV-driven lentiviral expression vectors encoding myc-XBP1s then infected with virus-containing supernatant from lytic iSLK.219 or TREx-BCBL1-RTA cultures the following day. After infection, the medium was replaced with fresh medium containing 1 μg/mL doxycycline. Supernatant was removed at 96 h post-infection and (A) the cell monolayer was fixed with paraformaldehyde and imaged. (B) Supernatant from *de novo* infected iSLK cells was harvested at 96 h post-infection and DNase I-protected viral genomic DNA was measured by qPCR as described above. The mean of four independent experiments is shown and the error bars correspond to the SEM. (****, p value < 0.0001) (C) Cell lysates were harvested at 96 h post-infection for immunoblot analysis for myc-epitope (XBP1s) and the viral proteins RTA (immediate early) and ORF57 (early). β-actin was used as a loading control. The immunoblots shown are representative of at least two independent experiments.

## Discussion

XBP1s-mediated transactivation of the RTA promoter causes KSHV reactivation from latency, but little is known about how ER stress and the UPR affect the ensuing lytic replication cycle. Here, we report that activation of all three UPR sensors (PERK, IRE1, ATF6) is required for

efficient KSHV lytic replication, because genetic or pharmacologic inhibition of each UPR sensor diminishes virion production. Despite UPR sensor activation during KSHV lytic replication, downstream UPR transcriptional responses were restricted. We do not yet know precisely how KSHV blocks the accumulation of ATF4 despite PERK activation and sustained eIF2α phosphorylation, nor do we understand why XBP1s protein does not accumulate despite IRE1 activation and *Xbp1* mRNA cleavage, or why ATF6(N) fails to transactivate target genes. The failure of SOX to suppress UPR transcription in our hands suggests that KSHV regulation of UPR is complex and likely represents the collective action of multiple viral gene products. Our ongoing studies are focused on identifying and characterizing these UPR-modulating viral gene products.

How might KSHV activate UPR sensors? All three UPR sensors are activated concurrently during the lytic cycle, which suggests that they may respond to canonical protein misfolding events in the ER. Such misfolding could result from biogenesis of viral glycoproteins, but we think this is unlikely because UPR sensor activation in the TREx BCBL1-RTA cell model precedes viral genome replication and bulk synthesis of structural proteins. We favour an alternative hypothesis whereby lytic replication disrupts ER chaperone function in some way, perhaps through the direct action of viral proteins or indirect activation of signal transduction pathways. Indeed, early in KSHV lytic replication, production of a viral G protein-coupled receptor (vGPCR) activates stress signalling and increases cytosolic calcium at least in part by inhibiting the SERCA calcium pump [85]. In this scenario, vGPCR may act similarly to Tg, inducing broad activation of all three UPR sensors by depleting ER calcium stores essential for protein folding. It seems counterintuitive that an enveloped virus would actively trigger ER stress, but we speculate that UPR activation in the earliest stages of lytic replication could help prime the ER for the impending burden of modifying secreted viral proteins and viral transmembrane glycoproteins. A more comprehensive accounting of changes in gene expression during early lytic replication is required to determine whether there is indeed an activation of an adaptive UPR to prepare the cell for late gene expression.

Why would a virus elicit ER stress but ultimately block the downstream UPR stress-mitigating transcription? UPR signalling is etiologically-linked to inflammatory diseases including Crohn's Disease and type 2 diabetes [86–88]. Furthermore, maximal IFN production following Toll-like receptor (TLR) engagement has been shown to require UPR activation and XBP1s activity [89]. Thus, KSHV suppression of UPR transcription may dampen antiviral inflammatory responses. Furthermore, because UPR transcription factors transactivate genes involved in catabolism, we speculate that blockade of their activity may allow newly synthesized viral gene products to evade degradation. For example, ATF6 and XBP1s transactivate genes involved in ERAD [29,61], so suppression of ATF6/XBP1s transcription could prevent ERAD-mediated degradation of newly-synthesized viral glycoproteins. Similarly, because ATF4 and its downstream target CHOP transactivate autophagy [16] and autophagy restricts herpesvirus replication [90,91], suppression of ATF4 and CHOP accumulation may allow KSHV to evade negative catabolic effects of autophagy. ATF4 and CHOP also promote apoptosis [92] and viral suppression of CHOP may thereby extend the survival of cells enduring the later stages of lytic replication.

The consequences of allowing unchecked accumulation of misfolded proteins during lytic replication remain unclear. Autophagy has been shown to compensate and assist with bulk protein degradation when ERAD is defective or inadequate for degrading misfolded proteins like aggregation-prone membrane proteins [93,94]. Since ERAD genes are not upregulated during lytic replication, it is possible that autophagy is required to clear misfolded proteins. However, there are multiple KSHV proteins that have been shown to directly inhibit autophagy including vFLIP, K-survivin, and vBCL2 [95–97], as well as multiple KSHV proteins that

stimulate the pro-growth PI3K-Akt-mTOR signalling pathway [98,99]. Simultaneous inhibition of ERAD and autophagy likely would cause the accumulation of the misfolded proteins, including aggregated membrane proteins, in KSHV infected cells. Mechanisms that KSHV employs (if any) to circumvent a possible accrual of misfolded proteins remain unknown.

The accumulating evidence indicates that herpesviruses have evolved distinct mechanisms to regulate the UPR to promote viral replication. For example, HSV-1 has been shown to dysregulate IRE1 activity [100]; HSV-1-infected cells displayed increased IRE1 protein levels but diminished IRE1 RNase activity, and treatment with the IRE1 inhibitor STF-083010 [101] diminished viral replication [102]. Moreover, HSV-1 glycoprotein B (gB) binds and inhibits PERK to support robust viral protein synthesis [56]. Interestingly, ectopic expression of FLAG-tagged XBP1s inhibits HSV-1 replication [102], consistent with our findings in the KSHV iSLK.219 infection model. By contrast, human cytomegalovirus (HCMV) lytic replication in fibroblast cells selectively activates PERK and IRE1, but not ATF6 [54]. Consistent with our model, IRE1 activation causes normal *Xbp1* mRNA splicing but fails to upregulate the XBP1s target gene EDEM (XBP1s protein levels were not evaluated in this study). However, unlike our observations in KSHV lytic infection, PERK causes ISR activation and normal ATF4 accumulation during HCMV lytic replication [54]. A recent study linked HCMV PERK and IRE1 activation to the UL148 glycoprotein [55]. In cells infected with a UL148-deficient HCMV, XBP1s protein levels were diminished compared to wildtype but levels of the XBP1s-target gene EDEM1 are unaffected, suggesting that XBP1s activity may also be inhibited in CMV-infected cells. Another study showed that HCMV UL38 induces eIF2α phosphorylation and ATF4 synthesis and promotes cell survival and virus replication during drug-induced ER stress by repressing JNK phosphorylation [103]. HCMV UL50 protein and the murine cytomegalovirus (MCMV) ortholog M50 bind IRE1 and promote its degradation, thereby inhibiting *Xbp1* mRNA splicing [104]. We suspect that many more viral regulators of the UPR will be discovered in the coming years. Such studies will benefit from the development of appropriate platforms to screen hundreds of viral ORFs for UPR and ISR-modulating activity.

The activation of IRE1 while simultaneously inhibiting XBP1s protein accumulation during KSHV lytic replication is curious. IRE1 can cleave a subset of ER-targeted mRNAs for degradation in a process called RIDD [31]. The physiological role of RIDD is unclear but it is thought that like the attenuation of translation by PERK, RIDD also helps to reduce the translation load in the ER. It is possible that certain viruses may hijack RIDD as a form of host shutoff to ensure preferential translation of secreted viral proteins. Conversely, if viral mRNAs contain XBP1-like cleavage sites, then the virus would likely want to suppress IRE1 RNase activity [105]. The molecular events that direct IRE1 toward RIDD or *Xbp1* mRNA splicing are not well understood but one study suggests that the higher-order oligomerization of IRE1 can dictate the response, whereby oligomeric IRE1 prefers *Xbp1* as a substrate, whereas dimeric IRE1 preferentially cleaves mRNAs through RIDD [106]. Since the levels of IRE1 are impacted by many of these viruses, including CMV, HSV-1, and KSHV, it will be interesting to determine if IRE1 specificity of mRNAs is impacted during infection.

We do not yet understand why XBP1s ectopic expression blocked KSHV virion production by iSLK.219 epithelial cells but not the PEL-derived (and engineered) TREx BCBL1-RTA cells. The rKSHV.219 (JSC-1 parental strain) and BCBL-1 strain are highly similar, with only significant differences evident in internal repeat regions and the K1 gene. We observed that these two virus strains, when used to infect naïve iSLK cells, were inhibited to a similar extent by ectopic XBP1s expression. This finding indicates that prior establishment of latency is not required to license XBP1s antiviral effects in epithelial cells, and also suggests that the failure of XBP1s to restrict KSHV virion production in the TREx BCBL1-RTA cell model may be due to cell intrinsic differences between epithelial cells and B lymphocytes. Additional mechanistic

studies will be required to fully explore these differences. Our work showed that ectopic XBP1s expression enabled RTA accumulation and KSHV reactivation from latency in both TREx BCBL1-RTA cells and iSLK.219 cells to a similar extent, and for the most part did not disrupt viral gene expression or genome replication. However, the diminished accumulation of structural proteins like ORF26 and ORF65 in iSLK.219 cells in the presence of high levels of XBP1s suggest a defect in the late stages of replication, which may give some clues about its antiviral mechanism of action. Thus, although XBP1s restriction of KSHV replication in the iSLK.219 cell model would seem to provide a tidy explanation for viral inhibition of XBP1s protein accumulation and UPR transcription, further mechanistic studies will be required to fully characterize the antiviral nature of XBP1s in certain cell types.

## Materials and methods

### Cell culture and chemicals

293A (ThermoFisher), HEK293T (ATCC), Hela Tet-Off (Clontech), iSLK and iSLK.219 cells (iSLK and iSLK.219 cells were kind gifts from Don Ganem) were cultured in DMEM (ThermoFisher) supplemented with 10% heat-inactivated fetal bovine serum (FBS), 100 Units/mL penicillin, 100 μg/mL streptomycin, and 2mM L-Glutamine. iSLK.219 cells were also passaged in the presence of 10 μg/mL of puromycin (ThermoFisher) to maintain the rKSHV.219 episomal DNA. TREx BCBL1-RTA cells (a kind gift from Jae Jung) were cultured in RPMI-1640 supplemented with 10% heat-inactivated FBS, 500 μM β-mercaptoethanol and the same concentrations of penicillin, streptomycin, and L-Glutamine as the adherent cell lines. All cells were maintained at 37˚C with 5% $CO_2$.

293A Tet-On cells were generated by transducing 293A cells with an MOI of < 1 of a lentiviral expression vector (see below for methods to generate lentiviral vectors and transducing cells) that encodes rtTA3 and the antibiotic selection marker for hygromycin B from the plasmid pLenti CMV rtTA3 Hygro (w785-1), which was a gift from Eric Campeau (Addgene plasmid # 26730). 24 h post-transduction fresh medium was added containing 200 μg/mL Hygromycin B (ThermoFisher) and positive transductants were selected for approximately 2 weeks.

To induce lytic replication via expression of the RTA transgene in TREx BCBL1-RTA and iSLK.219 cells, 1 μg/mL of doxycycline (dox; Sigma) was added to the cells. iSLK.219 cells were seeded at a density of $2x10^5$ cells/well of a 6-well plate one day prior to lytic induction and TREx BCBL1-RTA cells were seeded at a concentration of $2.5x10^5$ cells/mL immediately prior to lytic induction.

4μ8c (Axon Medchem and Sigma), PERKi (GSK2606414; Tocris), ISRIB (Sigma), Ceapin-A7 (Sigma), thapsigargin (Sigma), and tunicamycin (Sigma) were dissolved in DMSO (Sigma) and S1P inhibitor PF 429242 (Tocris) was dissolved in nuclease-free water to stock concentrations and diluted to the indicated concentrations in cell media.

### Plasmids

To generate the plasmids pLJM1 B* Puro and pLJM1 B* BSD, the genes encoding puromycin N-acetyltransferase and blasticidin S deaminase were amplified from pBMN-IRES-Puro and pBMN-IRES-Blast, respectively (previously generated in the lab by switching GFP with PuroR or BlastR from pBMN-I-GFP that was generated by the Nolan Lab [Stanford]), with forward and reverse primers containing BglII and KpnI RE sites. The antibiotic selection cassettes were cut-and-pasted into pLJM1.D (previously generated in the lab from modifying the MCS of pLJM1 plasmid that was generated by the Sabatini Lab [MIT]), deleting the BamHI RE site. A new MCS was generated by annealing overlapping oligos containing NheI and MfeI forward

and reverse RE sites and inserted into the plasmid digested with NheI and EcoRI to replace the existing MCS with unique RE sites in the following order: NheI, AgeI, BamHI, EcoRI, PstI, XbaI, MluI, SalI, EcoRV. 1x TO (tet operator) and 7x TO promoters were generated by annealing overlapping oligos containing one or seven copies of the tetracycline operator (5'-TCCCT ATCAGTGATAGAGA-3') and a minimal CMV promoter and using PCR extension to amplify a blunt oligo containing NdeI and NheI RE sites. The amplicon was digested with NdeI and NheI and pasted into the pLJM1 B* BSD replacing the CMV promoter.

To generate pLJM1 B* Puro HA-ATF6, HA-ATF6 was PCR amplified from pCGN-ATF6 (a gift from Ron Prywes, Addgene plasmid # 11974) and cut and pasted into pLJM1 B* Puro with NheI and AgeI.

pCMV2B-XBP1s was generated by PCR amplifying XBP1 from total RNA isolated from TREx BCBL1-RTA cells treated with Tg for 4h. After PCR, DNA was digested with PstI (cleaves unspliced XBP1 DNA only) to enrich for the spliced isoform of XBP1 and cut and pasted into pCMV2B with BamHI and XhoI and in-frame of the N-terminal FLAG-tag. pLJM1 BSD CMV, 1x TetO, and 7x TetO myc-XBP1s were generated by including the myc-tag nucleotide sequence in the 5' primer and amplifying XBP1s from pCMV2B-XBP1s. Cloned myc-XBP1s was subsequently cut and pasted into their corresponding lentiviral plasmids with BamHI and SalI. FLAG-RTA was cut from pcDNA3-FLAG-RTA (previously generated in the lab) with RE sites EcoRI and XhoI and pasted into pLJM1 B* BSD CMV and 7x TetO with RE sites EcoRI and SalI (via compatible sticky ends between XhoI and SalI).

pLJM1 BSD 7x TetO myc-SOX WT and P176S plasmids were generated by including the myc-tag nucleotide sequence in the 5' primer and amplifying wildtype SOX and the P176S point mutant from iSLK.219 and iSLK.BAC16 SOX P176S mutant genomic DNA [76], respectively. Amplified myc-SOX was cut and pasted into pLJM1 BSD 7x TetO via the RE sites NheI and SalI.

## Lentiviral vectors

HEK293T cells were seeded on 10 cm plates for 60–70% cell density the following day. Cells were transfected with polyethylenimine (PEI; Sigma) and the following plasmids for lentiviral generation: pLJM1 transfer plasmid, pMD2.G, and psPAX2. pMD2.G and psPAX2 are gifts from Didier Trono (Addgene plasmids # 12259 and # 12260). 48 h post-transfection, lentivirus containing supernatants were passed through a 0.45 µM filter and frozen at -80°C.

For transducing iSLK.219, iSLK.219 cells were seeded at a density of $5x10^4$ cells/well of a 6-well plate and the following day were resuspended in media containing 4 µg/mL polybrene (Sigma). Lentivirus was serially diluted dropwise onto cells and incubated at 37°C for 24 h. Following infection, the media was refreshed containing 10 µg/mL blasticidin. For *de novo* infection of iSLK cells, cells were seeded at a density of $2x10^5$ cells/well and transduced with a low dilution of lentivirus. These cells were infected one day after transduction and were not selected for lentivirus integration. For transducing TREx BCBL1-RTA cells, cells were seeded at a concentration of $2.5x10^5$ cells/mL in media containing 4 µg/mL polybrene and lentivirus was serially diluted onto cells. 24 h post-infection, the media was replaced with media containing 1 µg/mL puromycin. The first lentivirus dilution that resulted in minimal cell death following antibiotic selection was used for subsequent experiments.

## shRNA lentivirus cloning and knockdown

The pLKO.1 -TRC control plasmid was a gift from David Root (Addgene plasmid # 10879) and the RNAi Consortium was used to design the following oligos for generating shRNA lentiviral vectors against ATF6α and IRE1α with the targeting sequences underlined (Fwd/Rev):

shAFT6α (TRCN0000416318): 5'- CCGG<u>ACAGAGTCTCTCAGGTTAAAT</u>CTCGAGAT
TTAACCTGAGAGACTCTGTTTTTTG/5'- AATTCAAAAAACAGAGTCTCTCAGGTTA
AATCTCGAG<u>ATTTAACCTGAGAGACTCTGT</u>-3'

shIRE1α-1 (TRCN0000356305): 5'- CCGG<u>TCAACGCTGGATGGAAGTTTG</u>CTCGAGC
AAACTTCCATCCAGCGTTGATTTTTG -3'/5'- AATTCAAAAATCAACGCTGGATGGA
AGTTTGCTCGAG<u>CAAACTTCCATCCAGCGTTGA</u>-3'

shIRE1α-2 (TRCN0000235529): 5'- CCGG<u>AGAGGAGGGAATCGTACATTT</u>CTCGAGA
AATGTACGATTCCCTCCTCTTTTTTG-3'/5'- AATTCAAAAAAGAGGAGGGAATCGTA
CATTTCTCGAG<u>AAATGTACGATTCCCTCCTCT</u>-3'

The cloning strategy on Addgene (http://www.addgene.org/tools/protocols/plko/) was used to clone the oligos into either pLKO.1-Puro or pLKO.1-Blast and lentivirus generation was completed as described above.

## Fluorescent imaging

For Fig 8, iSLK.219 cells were seeded at a density of $2x10^5$ cells/well of a 6-well plate and 48 h post-dox, brightfield images and fluorescent images were captured with Olympus CKX41 microscope fitted with a QImaging QICAM Fast 1394 digital camera and Lumencor Mira light engine and using the 10x objective. For Fig 9, iSLK.219 cells were seeded at density of $5x10^4$ cells/well of a 6-well plate and transduced with the indicated lentiviral vectors. Fluorescent images were captured with an EVOS FL Cell Imaging System with 10x objective (ThermoFisher) following 5 days post-transduction or 48 h post-dox. Images of lytic foci (Fig 11A) were captured on an Axiovert 200M with a 5x objective using a Orca R2 monochrome camera Hamamatsu). All greyscale images were false coloured red or green for presentation.

## Immunoblotting

Cells were washed once with ice-cold PBS and lysed in 2x Laemmli Buffer (120mM Tris-HCl pH 6.8, 20% glycerol, 4% SDS). Lysates were passed through a 21-gauge needle 5–7 times to minimize viscosity and protein concentration was quantified with DC protein assay (Bio-Rad). DTT and bromophenol blue were added to samples, boiled, and 10–20 μg of protein per sample were loaded on 6–12% polyacrylamide gels and resolved by SDS-PAGE. Protein was transferred to PVDF membranes using the Trans-Blot Turbo Transfer System (Bio-Rad) and blocked in 5% skim milk diluted in Tris-buffered-saline (TBS) supplemented with 0.1% Tween 20 (Fisher Bio) (TBS-T) for 1 h at room temperature followed by overnight incubation at 4°C with primary antibody diluted in 5% bovine serum albumin in TBS-T. Following washing of primary antibody, membranes were incubated with IgG HRP-linked antibodies followed by exposure to ECL-2 western blotting substrate (Thermo Scientific, Pierce) and imaged by chemifluorescence on a Carestream Image Station 400mm Pro (Carestream) or using Clarity ECL luminescent reagent (Bio-Rad) and a ChemiDoc imaging system (Bio-Rad). The following antibodies were used: PERK (Cell Signaling Technology (CST); #5683); IRE1α (CST; #3294); ATF6α (Abcam; ab122897); XBP1 (CST; #12782); Phospho-eIF2α (Ser51; Abcam; ab32157); Total eIF2α (CST; #5324); ATF4 (Santa Cruz; sc-200); Myc-tag (CST; #2278); FLAG-tag (DYKDDDDK tag; CST; #8146); ORF26 (ThermoFisher; MA5-15742) ORF45 (ThermoFisher; MA5-14769); ORF57 (Santa Cruz; sc-135746); ORF65 (a kind gift from Shou-Jiang Gao); RTA (a kind gift from David Lukac); β-actin HRP conjugate (CST; #5125); anti-rabbit IgG HRP-linked (CST; #7074); anti-mouse IgG HRP-linked (CST; #7076).

## XBP1 RT-PCR splicing assay

RNA was isolated from TREx BCBL1-RTA cells with the RNeasy Plus Kit (Qiagen) and 500 ng total RNA was reverse transcribed with qScript cDNA SuperMix (Quanta) according to manufacturers' protocols. Based on the Ron Lab protocol (http://ron.cimr.cam.ac.uk/protocols/XBP-1.splicing.06.03.15.pdf) and minor modifications, a 473 bp PCR product overlapping the IRE1 splice site was amplified with XBP1 Fwd primer (5'-AAACAGAGTAGCAGCTCA-GACTGC-3') and XBP1 Rev primer (5'-TCCTTCTGGGTAGACCTCTGGGAG-3'). The amplified PCR product was digested overnight with 40 units of High Fidelity PstI (New England Biolabs) to cleave unspliced XBP1 cDNA. The PCR products were resolved on a 2.5% agarose gel made with 1x TAE (Tris-acetate-EDTA) buffer and stained with SYBR Safe (ThermoFisher) and visualized on a ChemiDoc Imaging Station (Bio-Rad). Percent *Xbp1* mRNA splicing was calculated by densitometry analysis with Image Lab software ver. 6.0.0 (Bio-Rad) and calculated using the following formula:

$$\% \textbf{ xbp1 mRNA splicing} = \frac{0.5 * \textbf{xbp1hybrid} + \textbf{xbp1s}}{\textbf{xbp1hybrid} + \textbf{xbp1s} + \textbf{xbp1u1} + \textbf{xbp1u2}} * 100$$

## Quantitative Reverse-Transcription PCR (RT-qPCR)

RNA was isolated from TREx BCBL1-RTA cells with the RNeasy Plus Kit (Qiagen) and 500 ng total RNA was reverse transcribed with qScript cDNA SuperMix (Quanta) according to manufacturers' protocols. A CFX96 Touch Real-Time PCR Detection System (Bio-Rad) and GoTaq qPCR MasterMix (Promega) was used to perform Real-Time PCR. Changes in mRNA levels were calculated by the $\Delta\Delta$Ct method [107] and normalized using 18S rRNA as a reference gene. The following primer sets were used in the study:

*18S* F: 5'-TTCGAACGTCTGCCCTATCAA-3'; R: 5'-GATGTGGTAGCCGTTTCTCAGG-3'
*ATF4*: 5'-CCACCATGGCGTATTAGGGG-3'; R: 5'-TAAATCGCTTCCCCCTTGGC-3'
*BiP* F: 5'-GCCTGTATTTCTAGACCTGCC-3'; R: 5'-TTCATCTTGCCAGCCAGTTG-3'
*CHOP*: 5'-ATGAACGGCTCAAGCAGGAA-3'; R: 5'-GGGAAAGGTGGGTAGTGTGG-3'
*EDEM1* F: 5'-TTGACAAAGATTCCACCGTCC-3'; R: 5'-TGTGAGCAGAAAG-GAGGCTTC-3'
*ERdj4* F: 5'-CGCCAAATCAAGAAGGCCT-3'; R: 5'-CAGCATCCGGGCTCTTATTTT-3'
*HERPUD1* F: 5'-AACGGCATGTTTTGCATCTG-3'; R: 5'-GGGGAAGAAAGGTTCCGA AG-3'
*K8.1* F: 5'-AGATACGTCTGCCTCTGGGT-3'; R: 5'-AAAGTCACGTGGGAGGTCAC-3'
*ORF26* F: 5'-CAGTTGAGCGTCCCAGATGA-3'; R: 5'-GGAATACCAACAGGAGGCCG-3'
*ORF45* F: 5'-TGATGAAATCGAGTGGGCGG-3'; R: 5'-CTTAAGCCGCAAAGCAGTGG-3'
*RTA* F: 5'-GATTACTGCGACAACGGTGC-3'; R: 5'-TCTGCGACAAAACATGCAGC

## Viral genome amplification

DNA was harvested from iSLK.219 cells with DNeasy Blood & Tissue Kit (Qiagen) according to manufacturer's protocol. RT-PCR was carried out as described previously with primers specific to KSHV ORF26 (as listed previously) and β-actin (F: 5'-CTTCCAGCAGATGTGGAT CA-3'; R: 5'-AAAGCCATGCCAATCTCATC-3'). Changes in KSHV genome copy number was calculated by the $\Delta\Delta$Ct method and normalized to β-actin.

## Titering DNAse-protected viral genomes

Virus containing cell supernatants were processed by first pelleting floating cells and debris and then 180 μL cleared supernatant were treated with 20 μL of 3 mg/mL DNase I (Sigma) for

30 minutes at 37˚C. Viral genomic DNA was then purified from the supernatant with DNeasy Blood & Tissue Kit (Qiagen) according to the manufacturer's protocol with the following modifications: 10 μg of salmon sperm DNA (Invitrogen) and 1 ng of luciferase plasmid pGL4.26 [*luc2*/minP/Hygro] (Promega) were added to the lysis buffer. RT-PCR was performed as described previously with primers specific to KSHV ORF26 (as listed previously) and *luc2* (F: 5'-TTCGGCAACCAGATCATCCC-3'; R: 5'-TGCGCAAGAATAGCTCCTCC-3'). Changes in virus titer was calculated by the ΔΔCt method and normalized to *luc2*.

## rKSHV.219 infection and titering

Virus-containing supernatant was harvested from iSLK.219 cells at the indicated times by pelleting cellular debris at 3300 x g for 5 minutes and then stored at -80˚C until ready to titer the virus. $10^5$ 293A cells/well were seeded in 12-well plates to obtain a confluent monolayer two days later. The thawed viral inoculum was briefly vortexed and centrifuged again at 3300 x g for 5 minutes. Two-fold serial dilutions of viral supernatants were applied to the monolayer containing 4 μg/mL polybrene and 25 mM HEPES (Gibco) and centrifuged at 800 x g for 2 h at 30˚C. The total cell count per well was also determined from an uninfected well. Fresh media was applied immediately after spinoculation. 20–24 h post-infection, two dilutions that resulted in less than 30% GFP-positive cells (the linear range for infection) were trypsinized, washed once with PBS and fixed with 1% paraformaldehyde in PBS. GFP-positivity was measured on either FACSCalibur or FACSCanto cytometers (BD) by gating on FSC/SSC and counting 10000–15000 "live" events. Gating and % GFP positive events were determined with FCS Express 6 Flow Cytometry Software (ver.6.0; De Novo). Virus titer was calculated as IU/mL with the following formula:

$$\textbf{Virus titre} \left( \frac{\textbf{IU}}{\textbf{mL}} \right) = \% \textbf{ GFP positive events} * \textbf{dilution factor} * \textbf{cell count}/100$$

The virus titer of the two dilutions were averaged for the final titer value.

## *De novo* infection

rKSHV.219 was harvested from the supernatant of iSLK.219 cells four days after lytic reactivation with 1 μg/mL doxycycline. Virus from TREx-BCBL1-RTA cells was isolated from cultures 48 h after reactivation. Cell debris was removed, and aliquots of supernatant were processed and stored as described above. $2x10^5$ iSLK cells/well were seeded in 6 well plates and the following day cells were transduced with lentiviral vectors as described above. The next day a 1:10 dilution of viral supernatant supplemented with 4 μg/mL polybrene was added to the cells, which were then centrifuged as described above. The inoculum was removed after spinoculation and medium was replaced with fresh media, or media containing 1 μg/mL doxycycline. Supernatant from the *de novo* infected cells were harvested and analysed for DNase I-protected genomes as described above.

## Statistical analysis

Prism7 (GraphPad) was used for generating graphs and performing statistical analysis. Unpaired Student's t-tests were used to determine significance between two groups. One-way or two-way ANOVA was used to compare multiple samples or between grouped samples respectively, and an appropriate post-hoc test was done to determine differences between groups. p-values <0.05 were considered significant and denoted as the following: <0.05 (*), <0.01 (**), <0.001 (***), <0.0001 (****)

## Acknowledgments

We thank members of the McCormick lab for critical reading of this manuscript. We would like to thank Don Ganem (UCSF), Shou-Jiang Gao (USC), Jae Jung (USC), and David Lukac (Rutgers) for providing reagents.

## Author Contributions

**Conceptualization:** Benjamin P. Johnston, Craig McCormick.

**Funding acquisition:** Craig McCormick.

**Investigation:** Benjamin P. Johnston, Eric S. Pringle, Craig McCormick.

**Methodology:** Benjamin P. Johnston, Eric S. Pringle.

**Project administration:** Craig McCormick.

**Supervision:** Craig McCormick.

**Writing – original draft:** Benjamin P. Johnston, Craig McCormick.

**Writing – review & editing:** Benjamin P. Johnston, Craig McCormick.

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
