## [Decision Letter · Decision Letter 0]

8 Aug 2019

Dear Dr. McCormick,

Thank you very much for submitting your manuscript "KSHV activates unfolded protein response sensors but suppresses downstream transcriptional responses to support lytic replication" (PPATHOGENS-D-19-01170) for review by PLOS Pathogens. Your manuscript was fully evaluated at the editorial level and by independent peer reviewers. The reviewers appreciated the attention to an important problem, but raised some substantial concerns about the manuscript as it currently stands. These issues must be addressed before we would be willing to consider a revised version of your study. We cannot, of course, promise publication at that time.

We therefore ask you to modify the manuscript according to the review recommendations before we can consider your manuscript for acceptance. Your revisions should address the specific points made by each reviewer.

(1) A letter containing a detailed list of your responses to the review comments and a description of the changes you have made in the manuscript. Please note while forming your response, if your article is accepted, you may have the opportunity to make the peer review history publicly available. The record will include editor decision letters (with reviews) and your responses to reviewer comments. If eligible, we will contact you to opt in or out.

(2) Two versions of the manuscript: one with either highlights or tracked changes denoting where the text has been changed; the other a clean version (uploaded as the manuscript file).

Additionally, to enhance the reproducibility of your results, PLOS recommends that you deposit your laboratory protocols in protocols.io, where a protocol can be assigned its own identifier (DOI) such that it can be cited independently in the future. For instructions see http://journals.plos.org/plospathogens/s/submission-guidelines#loc-materials-and-methods

We hope to receive your revised manuscript within 60 days. If you anticipate any delay in its return, we ask that you let us know the expected resubmission date by replying to this email. Revised manuscripts received beyond 60 days may require evaluation and peer review similar to that applied to newly submitted manuscripts.

[LINK]

Sincerely,

Sankar Swaminathan, MD

Associate Editor

PLOS Pathogens

Shou-Jiang Gao

Section Editor

PLOS Pathogens

Kasturi Haldar

Editor-in-Chief

PLOS Pathogens

orcid.org/0000-0001-5065-158X

Grant McFadden

Editor-in-Chief

PLOS Pathogens

orcid.org/0000-0002-2556-3526

While the reviewers appreciated the thorough responses to the prior review, it was pointed out that the new experiments did not confirm the observed UPR related effects in another cell type. The reviewer points out that it is not clear whether the UPR effects are a virus-specific phenomenon and it cannot be concluded that this is a cell dependent variability. It would seem reasonable to attempt to show whether this is a virus-strain or cell based difference. While it might be more difficult to generate a B lymphocyte cell line with the recombinant virus, it should be possible to examine the phenotype of the recombinant virus in another epithelial cell line.

Reviewer's Responses to Questions

**Part I - Summary**

Reviewer #1: In this revised manuscript Johnston & McCormick characterize the kinetics of UPR activation during KSHV reactivation in iSLK and Trex-BCBL1 cells. They nicely show activation of UPR as evidenced by ATF6 cleavage, PERK phosphorylation, and Xbp1 splicing, but reveal that subsequent UPR-triggered transcriptional responses and XBP1s protein accumulation do not occur. Unexpectedly, overexpression of XBP1s appears to have opposing outcomes in iSLK versus the BCBL1 model, but knockdown or pharmacological inhibition of UPR prior to reactivation suppresses late stage events of the viral lifecycle in both cell types. This leads them to hypothesize that components of the UPR are being repurposed for viral benefit. Although the mechanism by which outputs of the UPR are being modulated by the virus remains unknown, this study nicely characterizes the UPR activation landscape during KSHV replication. The resubmission is improved through inclusion of data suggesting that the blunting of downstream UPR responses is not due to host shutoff and through new data bolstering the conclusion that KSHV inhibits the ISR response.

Reviewer #2: In this revision of their manuscript the authors have addressed the majority of the points raised by the reviewers. However, in addressing the issue that they only show inhibition of KSHV replication by XBP1s in iSLK cells they found that this did not hold true in the BCBL-1 derived cells. They conclude that this is due to cell type differences. However, two issues with this conclusion arise. First, the iSLK.219 virus is derived from JSC-1 cells and is heavily modified through introduction of multiple changes to the viral genome while the BCBL-1 derived virus is from a different primary effusion lymphoma sample that does not reactivate as well as the JSC-1 derived virus and is not a modified recombinant virus. Therefore, the differences in the ability of XBP1s to inhibit replication could be virus strain specific or due to the engineered alterations to the viral genome rather than cell type specificity. Secondly, the effect of XBP1s does not occur in the more relevant B-cell only in the highly transformed iSLK cells. Therefore, both viruses should be used in a relevant cell type to determine the importance of the inhibition of replication by XBP1s.

**Part II – Major Issues: Key Experiments Required for Acceptance**

Reviewer #1: none

Reviewer #2: (No Response)

**Part III – Minor Issues: Editorial and Data Presentation Modifications**

Reviewer #1: none

Reviewer #2: (No Response)

PLOS authors have the option to publish the peer review history of their article (what does this mean?). If published, this will include your full peer review and any attached files.

Reviewer #1: No

Reviewer #2: No

---

## [Editor Report · Decision Letter 1]

2 Nov 2019

Dear Dr. McCormick,

We are pleased to inform that your manuscript, "KSHV activates unfolded protein response sensors but suppresses downstream transcriptional responses to support lytic replication", has been editorially accepted for publication at PLOS Pathogens. 

Before your manuscript can be formally accepted and sent to production, you will need to complete our formatting changes, which you will receive by email within a week. Please note that your manuscript will not be scheduled for publication until you have made the required changes.

IMPORTANT NOTES

(1) Please note, once your paper is accepted, an uncorrected proof of your manuscript will be published online ahead of the final version, unless you’ve already opted out via the online submission form. If, for any reason, you do not want an earlier version of your manuscript published online or are unsure if you have already indicated as such, please let the journal staff know immediately at plospathogens@plos.org.

(2) Copyediting and Proofreading: The corresponding author will receive a typeset proof for review, to ensure errors have not been introduced during production. Please review the PDF proof of your manuscript carefully, as this is the last chance to correct any errors. Please note that major changes, or those which affect the scientific understanding of the work, will likely cause delays to the publication date of your manuscript. 

(3) Appropriate Figure Files: Please remove all name and figure # text from your figure files. Please also take this time to check that your figures are of high resolution, which will improve the readbility of your figures and help expedite your manuscript's publication. Please note that figures must have been originally created at 300dpi or higher. Do not manually increase the resolution of your files. For instructions on how to properly obtain high quality images, please review our Figure Guidelines, with examples at: http://journals.plos.org/plospathogens/s/figures.

(4) Striking Image: Please upload a striking still image to accompany your article if one is available (you can include a new image or an existing one from within your manuscript). Should your paper be accepted, this image will be considered for our monthly issue image and may also appear on our website to feature your article. Please upload this as a separate file, selecting "striking image" as the file type upon upload. Please also include a separate "Other" file with a caption, including credits and any potential copyright information. Please do not include the caption in the main article file. If your image is from someone other than yourself, please ensure that the artist has read and agreed to the terms and conditions of the Creative Commons Attribution License at http://journals.plos.org/plospathogens/s/content-license. Please note that PLOS cannot publish copyrighted images.

(5) Press Release or Related Media: If your institution or institutions have a press office, please notify them about your upcoming paper at this point, to enable them to help maximize its impact. If they will be preparing press materials for this manuscript, please inform our press team in advance at plospathogens@plos.org as soon as possible. We ask that you contact us within one week to plan ahead of our fast Production schedule. If you need to know your paper's publication date for related media purposes, you must coordinate with our press team, and your manuscript will remain under a strict press embargo until the publication date and time. This means an early version of your manuscript will not be published ahead of your final version. 

(6)  PLOS requires an ORCID iD for all corresponding authors on papers submitted after December 6th, 2016. Please ensure that you have an ORCID iD and that it is validated in Editorial Manager.  To do this, go to ‘Update my Information’ (in the upper left-hand corner of the main menu), and click on the Fetch/Validate link next to the ORCID field.  This will take you to the ORCID site and allow you to create a new iD or authenticate a pre-existing iD in Editorial Manager

(7) Update your Profile Information: Now that your manuscript has been provisionally accepted, please log into Editorial Manager and update your profile, if needed. Go to https://www.editorialmanager.com/ppathogens, log in, and click on the "Update My Information" link at the top of the page. Please update your user information to ensure an efficient production and billing process. 

(8) LaTeX users only: Our staff will ask you to upload a TEX file in addition to the PDF before the paper can be sent to typesetting, so please carefully review our Latex Guidelines http://journals.plos.org/plospathogens/s/latex in the meantime.

(9) If you have associated protocols in protocols.io, please ensure that you make them public before publication to guarantee immediate access to the methodological details.

Best regards,

Sankar Swaminathan, MD

Associate Editor

PLOS Pathogens

Shou-Jiang Gao

Section Editor

PLOS Pathogens

Kasturi Haldar

Editor-in-Chief

PLOS Pathogens

orcid.org/0000-0001-5065-158X

Grant McFadden

Editor-in-Chief

PLOS Pathogens

orcid.org/0000-0002-2556-3526

The response and revised manuscript address all the questions raised by the reviewers.
---

## [Editor Report · Acceptance letter]

26 Nov 2019

Dear Dr. McCormick,

We are delighted to inform you that your manuscript, "KSHV activates unfolded protein response sensors but suppresses downstream transcriptional responses to support lytic replication," has been formally accepted for publication in PLOS Pathogens.

Best regards,

Kasturi Haldar

Editor-in-Chief

PLOS Pathogens

orcid.org/0000-0001-5065-158X

Grant McFadden

Editor-in-Chief

PLOS Pathogens

orcid.org/0000-0002-2556-3526